# Generative Neural Fields
# by Mixtures of Neural Implicit Functions

**Tackgeun You**[3]
tackgeun.you@postech.ac.kr

**Mijeong Kim**[1]
mijeong.kim@snu.ac.kr

**Jungtaek Kim**[4]
jungtaek.kim@pitt.edu

**Bohyung Han**[1,2]
bhhan@snu.ac.kr

[1]ECE & [2]IPAI, Seoul National University, South Korea
[3]CSE, POSTECH, South Korea
[4]University of Pittsburgh, USA

## Abstract

We propose a novel approach to learning the generative neural fields represented by linear combinations of implicit basis networks. Our algorithm learns basis networks in the form of implicit neural representations and their coefficients in a latent space by either conducting meta-learning or adopting auto-decoding paradigms. The proposed method easily enlarges the capacity of generative neural fields by increasing the number of basis networks while maintaining the size of a network for inference to be small through their weighted model averaging. Consequently, sampling instances using the model is efficient in terms of latency and memory footprint. Moreover, we customize denoising diffusion probabilistic model for a target task to sample latent mixture coefficients, which allows our final model to generate unseen data effectively. Experiments show that our approach achieves competitive generation performance on diverse benchmarks for images, voxel data, and NeRF scenes without sophisticated designs for specific modalities and domains.

## 1 Introduction

Implicit neural representation (INR) is a powerful and versatile tool for modeling complex and diverse data signals in various modalities and domains, including audio [12], images [35], videos [40], 3D objects [28, 6], and natural scenes. INR expresses a data instance as a function mapping from a continuous coordinate space to a signal magnitude space rather than using a conventional representation on a discrete structured space. In particular, a representation with a continuous coordinate allows us to query arbitrary points and retrieve their values, which is desirable for many applications that only have accessibility to a subset of the target data in a limited resolution. INRs replace the representations based on high-dimensional regular grids, such as videos [9] and 3D scenes [25], with multi-layer perceptrons (MLPs) with a relatively small number of parameters, which effectively memorize scenes.

Generative neural fields aim to learn distributions of functions that represent data instances as a form of neural field. They typically rely on INRs to sample their instances in various data modalities and domains. The crux of generative neural fields lies in how to effectively identify and model shared and instance-specific information. To this end, feature-wise linear modulation (FiLM) [29] and hyper-network (HyperNet) [16] are common methods for modulating representations. FiLM

37th Conference on Neural Information Processing Systems (NeurIPS 2023).

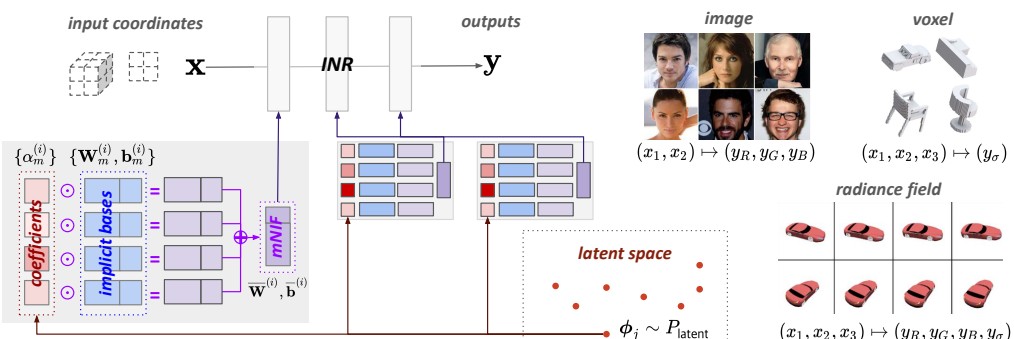

Figure 1: Overview of the generation procedure using the proposed generative neural field based on mixtures of neural implicit function (mNIF). Our model is applicable to various types of data such as images, voxels, and radiance fields. To generate an instance, we first sample a context vector $\phi_j$ from a prior distribution ($P_{\text{latent}}$) estimated by a denoising diffusion probabilistic model. We then perform a weighted model averaging on implicit bases $\{\mathbf{W}_m^{(i)} \mathbf{b}_m^{(i)}\}$ using mixture coefficients ($\alpha_m^{(i)}$) derived from the context vector.

adjusts hidden features using an affine transform, which consists of element-wise multiplication and bias addition. Due to the ease of optimization, the effectiveness of FiLM has already been validated in several tasks including 3D-aware generative modeling [6, 7]. However, the operation of FiLM is too restrictive while introducing additional parameters for modulation. On the other hand, HyperNet is more flexible because it directly predicts network parameters. Unfortunately, the direct prediction of model parameters is prone to unstable training and limited coverage of the data distribution generated by the predicted network. Therefore, to reduce solution spaces, HyperNet often employs dimensionality reduction techniques such as low-rank decompositions of weight matrices [35, 12] and combinations of multiple network blocks [16]. However, all these works give limited consideration to the inference efficiency.

Inspired by the flexibility of HyperNet and the stability of FiLM while considering efficiency for inference, we propose a *mixture of neural implicit functions* (mNIF) to represent generative neural fields. Our approach employs mNIFs to construct generative networks via model averaging of INRs. The mixture components in an mNIF serve as shared implicit basis networks while their mixands define relative weights of the bases to construct a single instance. Our modulation step corresponds to computing a weighted average of the neural implicit functions. The mixture coefficients are optimized by either meta-learning or auto-decoding procedures with signal reconstruction loss. Such a design is effective for maximizing the expressibility of INRs while maintaining the compactness of inference networks. The proposed approach is versatile for diverse modalities and domains and is easy to implement.

Figure 1 demonstrates the overview of the proposed approach, and we summarize the contributions of our work as follows:

- We introduce a generative neural field based on mNIF, a modulation technique by a linear combination of NIFs. The proposed approach effectively extracts instance-specific information through conducting meta-learning or applying auto-decoding procedures.

- Our modulation allows us to easily extend the model capacity by increasing the number of basis networks without affecting the structure and size of the final model for inference. Such a property greatly improves the efficiency of the sampling network in terms of inference time and memory requirements.

- Our model achieves competitive performance on several data generation benchmarks without sophisticated domain-specific designs of algorithm and architecture configuration.

The rest of this paper is organized as follows. We first discuss related works about generative neural fields in Section 2. The main idea of the proposed approach and the training and inference procedures are discussed in Sections 3 and 4, respectively. Section 5 presents empirical quantitative and qualitative results with discussions, and Section 6 concludes this paper.

## 2 Related Work

Implicit neural representations (INR) are often optimized for representing a single instance in a dataset, and have been extended in several ways to changing activation functions [33, 30], utilizing input coordinate embeddings [36, 26] or taking advantage of a mixture of experts [37, 23, 24]. Since the INR framework is difficult to generalize a whole dataset, generative neural fields have been proposed to learn a distribution of functions based on INRs, each of which corresponds to an example in the dataset.

Generative manifold learning (GEM) [12] and generative adversarial stochastic process (GASP) [14] are early approaches to realizing generative neural fields. GEM implements the generative neural field using the concept of hyper-network (HyperNet) [16], which predicts a neural network representing an instance. Training GEM is given by the auto-decoding paradigm with manifold learning. However, GEM involves extra computational cost because the manifold constraint requires accessing training examples. GASP also employs a HyperNet for sampling instances, and its training utilizes a discriminator taking an input as a set of coordinate and feature pairs. Since the output of the discriminator should be permutation invariant, it follows the design of the network for the classification of point cloud data. However, training GASP is unstable and the discriminator design is sometimes tricky, especially in NeRF. Functa [13] proposes the generative neural fields with SIREN [33] modulated by FiLM [29] and introduces a training strategy based on meta-learning. However, meta-learning is computationally expensive due to the Hessian computation. On the other hand, diffusion probabilistic fields (DPF) [41] proposes a single-stage diffusion process with the explicit field parametrization [19]. However, its high computational cost hampers the applicability to complex neural fields such as NeRF. HyperDiffusion [15], which is concurrent to our work, presents a framework directly sampling the entire INR weight from the learned diffusion process instead of exploiting latent vectors for modulating INR weights. However, this work demonstrates instance generation capability only on voxel domains, not on images.

Model averaging [38] is often used for enhancing performance in discriminative tasks without increasing an inference cost. Our approach applies model averaging to generative models and verifies its effectiveness in generative neural fields based on INRs. Note that we are interested in efficient prediction by learning a compact representation through a weighted mixture; our method jointly trains basis models and their mixture coefficients.

## 3 Generative Neural Fields with a Mixture of Neural Implicit Functions

This section describes how to define generative neural fields. The formulation with a mixture of neural implicit functions (mNIF) enforces implicit basis networks to represent shared information across examples and allows latent mixture coefficients to encode instance-specific information through a linear combination of basis networks. We show that generative neural field via mNIF is effectively optimized to predict the latent vector for the construction of models generating high-quality samples.

### 3.1 Implicit Neural Representations

Implicit neural representation (INR) expresses a data instance using a function from an input query, $\mathbf{x} \in \mathbb{N}^d$, to its corresponding target value, $\mathbf{y} \in \mathbb{R}^k$. Since each input coordinate is independent of the others, a mapping function parametrized by $\boldsymbol{\theta}$, $\mathbf{f}_{\boldsymbol{\theta}}(\cdot)$, is typically represented by a multi-layer perceptron (MLP) with $L$ fully connected (FC) layers, which is given by

$$\mathbf{y} = \mathbf{f}_{\boldsymbol{\theta}}(\mathbf{x}) = \mathbf{f}^{(L+1)} \circ \cdots \circ \mathbf{f}^{(1)} \circ \mathbf{f}^{(0)}(\mathbf{x}), \tag{1}$$

where $\mathbf{f}^{(i)}$ for $\forall i \in \{1, 2, \ldots, L\}$ denote FC layers while $\mathbf{f}^{(0)}$ and $\mathbf{f}^{(L+1)}$ indicate the input and output layers, respectively. Each layer $\mathbf{f}^{(i)}$ performs a linear transform on its input hidden state $\mathbf{h}^{(i)}$ and then applies a non-linear activation function to yield the output state $\mathbf{h}^{(i+1)}$, expressed as

$$\mathbf{h}^{(i+1)} = \mathbf{f}^{(i)}(\mathbf{h}^{(i)}) = \sigma\left(\mathbf{W}^{(i)}\mathbf{h}^{(i)} + \mathbf{b}^{(i)}\right), \ i \in \{1, 2, \ldots, L\}, \tag{2}$$

where $\sigma(\cdot)$ is an activation function and $\mathbf{W}^{(i)} \in \mathbb{R}^{W \times W}$ and $\mathbf{b}^{(i)} \in \mathbb{R}^W$ are learnable parameters. The operations of the input and output layers are defined as $\mathbf{h}_1 = \mathbf{f}^{(0)}(\mathbf{x})$ and $\hat{\mathbf{y}} = \mathbf{f}^{(L+1)}(\mathbf{h}^{(L)})$,

respectively. Consequently, a collection of learnable parameters $\boldsymbol{\theta}$ in all layers is given by

$$\boldsymbol{\theta} \triangleq \left\{ \mathbf{W}^{(0)}, \mathbf{b}^{(0)}, \mathbf{W}^{(1)}, \mathbf{b}^{(1)}, \cdots, \mathbf{W}^{(L+1)}, \mathbf{b}^{(L+1)} \right\}, \tag{3}$$

where $\mathbf{W}^{(0)} \in \mathbb{R}^{W \times d}$, $\mathbf{b}^{(0)} \in \mathbb{R}^{W}$, $\mathbf{W}^{(L+1)} \in \mathbb{R}^{k \times W}$, and $\mathbf{b}^{(L+1)} \in \mathbb{R}^{k}$. For INR, the mean squared error between a prediction $\hat{\mathbf{y}}$ and a target $\mathbf{y}$ is typically adopted as a loss function, $\mathcal{L}(\cdot, \cdot)$, which is given by

$$\mathcal{L}(\hat{\mathbf{y}}, \mathbf{y}) = ||\hat{\mathbf{y}} - \mathbf{y}||^2. \tag{4}$$

Among the variations of INRs, we employ SIREN [33], which adopts the sinusoidal function as an activation function and introduces a sophisticated weight initialization scheme for MLPs. Supposing that SIREN is defined with an MLP with $L$ hidden layers, the $i^{\text{th}}$ layer of SIREN is given by

$$\mathbf{h}^{(i+1)} = \sin \left( w_0 \left( \mathbf{W}^{(i)} \mathbf{h}^{(i)} + \mathbf{b}^{(i)} \right) \right), \ i \in \{1, \ldots, L\}, \tag{5}$$

where $w_0$ is a scale hyperparameter to control the frequency characteristics of the network. The initialization of SIREN encourages the distribution of hidden sine activations to follow a normal distribution with a standard deviation of 1.

## 3.2 Mixtures of Neural Implicit Functions

We propose generative neural fields based on mNIF, which is an extension of the standard INR for modulating its model weight. We define a set of NIFs, which is used as basis networks, and construct a generative neural field using a mixture of the NIFs. This is motivated by our observation that a generative neural field is successfully interpolated using multiple basis networks and model averaging works well for model generalization [38].

The operation in each layer of our mixture model is given by

$$\mathbf{h}^{(i+1)} = \sin \left( w_0 \left( \sum_{m=1}^{M} \alpha_m^{(i)} \mathbf{g}_m(\mathbf{h}^{(i)}) \right) \right), \tag{6}$$

where $\mathbf{g}_m^{(i)}(\mathbf{h}^{(i)}) = \mathbf{W}_m^{(i)} \mathbf{h}^{(i)} + \mathbf{b}_m^{(i)}$ is the $i^{\text{th}}$-layer operation of the $m^{\text{th}}$ neural implicit function, and $\alpha_m^{(i)}$ is a mixture coefficient of the same layer of the same neural implicit function. Note that modulating the network is achieved by setting the mixand values, $\{\alpha_m^{(i)}\}$. Similar to the definition of $\boldsymbol{\theta}$ described in Eq. (3), the parameter of each mixture is given by

$$\boldsymbol{\theta}_m \triangleq \left\{ \mathbf{W}_m^{(0)}, \mathbf{b}_m^{(0)}, \cdots, \mathbf{W}_m^{(L+1)}, \mathbf{b}_m^{(L+1)} \right\}. \tag{7}$$

The operation in the resulting INR obtained from model averaging is given by

$$\sum_{m=1}^{M} \alpha_m^{(i)} \mathbf{g}_m(\mathbf{h}^{(i)}) = \left( \sum_{m=1}^{M} \alpha_m^{(i)} \mathbf{W}_m^{(i)} \right) \mathbf{h}^{(i)} + \sum_{m=1}^{M} \alpha_m^{(i)} \mathbf{b}_m^{(i)} = \overline{\mathbf{W}}^{(i)} \mathbf{h}^{(i)} + \overline{\mathbf{b}}^{(i)}, \tag{8}$$

where

$$\overline{\mathbf{W}}^{(i)} \triangleq \sum_{m=1}^{M} \alpha_m^{(i)} \mathbf{W}_m^{(i)} \quad \text{and} \quad \overline{\mathbf{b}}^{(i)} \triangleq \sum_{m=1}^{M} \alpha_m^{(i)} \mathbf{b}_m^{(i)}. \tag{9}$$

All the learnable parameters in the mixture of NIFs is defined by $\{\boldsymbol{\theta}_1, \ldots, \boldsymbol{\theta}_M\}$. The remaining parameters are mixture coefficients, and we have the following two options to define them: (i) sharing mixands for all layers $\alpha_m^{(i)} = \alpha_m$ and (ii) setting mixands to different values across layers. The first option is too restrictive for the construction of INRs because all layers share mixture coefficients while the second is more flexible but less stable because it involves more parameters and fails to consider the dependency between the coefficients in different layers. Hence, we choose the second option but estimate the mixture coefficients in a latent space, where our method sets the dimensionality of the latent space to a constant and enjoys the high degree-of-freedom of layerwise coefficient setting. To this end, we introduce a projection matrix $\mathbf{T}$ to determine the mixture coefficients efficiently and effectively, which is given by

$$\left[ \alpha_1^{(0)}, \cdots, \alpha_M^{(0)}, \alpha_1^{(1)}, \cdots, \alpha_M^{(1)}, \cdots, \alpha_1^{(L+1)}, \cdots, \alpha_M^{(L+1)} \right] = \mathbf{T}\boldsymbol{\phi}, \tag{10}$$

where $\phi \in \mathbb{R}^H$ denotes a latent mixture coefficient vector. This choice compromises the pros and cons of the two methods. Eventually, we need to optimize a set of parameters, $\theta_{\text{all}} \triangleq \{\theta_1, \ldots, \theta_M, \mathbf{T}\}$ for a generative neural field, and a latent vector $\phi$ to obtain an instance-specific INR.

## 3.3 Model Efficiency

The proposed modulation technique is based on model averaging and has great advantages in the efficiency during training and inference. First, the computational complexity for each input coordinate is small despite a large model capacity given by potentially many basis networks. Note that, even though the number of neural implicit basis function in a mixture increases, the computational cost per query is invariant by Eq. (8), which allows us to handle more coordinates queries than the models relying on FiLM [29]. Second, the model size of our generative neural fields remains small compared to other methods due to our model averaging strategy, which is effective for saving memory to store the neural fields.

When analyzing the efficiency of generative neural fields, we consider two aspects; one is the model size in the context adaptation stage, in other words, the number of all the learnable parameters, and the other is the model size of the modulated network used for inference. By taking advantage of the modulation via model averaging, our approach reduces memory requirement significantly both in training and inference.

## 4 Learning Generative Neural Fields

The training procedure of the proposed approach is composed of two stages: context adaptation followed by task-specific generalization. In the context adaptation stage, we optimize all the learnable parameters for implicit basis functions and their coefficients that minimize reconstruction error on examples in each dataset. Then, we further learn to enhance the generalization performance of the task-specific generative model, where a more advanced sampling method is employed to estimate a context vector, *i.e.*, a latent mixture coefficient vector. This two-stage approach enables us to acquire the compact representation for neural fields and saves computational costs during training by decoupling the learning neural fields and the training generative model. We describe the details of the two-stage model below.

### 4.1 Stage 1: Training for Context Adaptation

The context adaptation stage aims to learn the basis networks and the latent mixture coefficient vector, where the mixture model minimizes the reconstruction loss. This optimization is achieved by one of two approaches: meta-learning or auto-decoding paradigm.

With meta-learning, we train models in a similar way as the fast context adaptation via meta-learning (CAVIA) [42]; after a random initialization of a latent vector $\phi_j$ for the $j^{\text{th}}$ instance in a batch, we update $\phi_j$ in the inner-loop and then adapt the shared parameters $\theta$ based on the updated $\phi_j$. Since this procedure requires estimating the second-order information for the gradient computation of $\theta$, the computational cost of this algorithm is high. Algorithm 1 presents this meta-learning procedure.

We observe that random initialization of a latent vector during meta-learning is more stable than the constant initialization originally proposed in CAVIA when training mNIF with a large number of mixture components or the high dimensionality of latent vectors. We intend to diversify INR weights at the start phase of meta-training, ensuring that not all samples are represented with identical INR weights.

Auto-decoding paradigm [12, 28] is a variant of multi-task learning, where each instance is considered a task. Contrary to the meta-learning, we do not reset and initialize the latent vector $\phi_j$ for each instance but continue to update over multiple instances. The optimization procedure via auto-decoding is presented in Algorithm 2.

### 4.2 Stage 2: Optimizing for Task-Specific Generalization

The aforementioned context adaptation procedure focuses only on the minimization of reconstruction error for training samples and may not be sufficient for the generation of unseen examples. Hence,

---

**Algorithm 1** Meta-learning with mNIF

1: Randomly initialize the shared parameter $\boldsymbol{\theta}$ of mNIF.
2: **while** training **do**
3:      Sample a mini-batch $\mathcal{B} = \{(\mathbf{c}_j, \mathbf{y}_j)\}_{j=1:|\mathcal{B}|}$.
4:      **for** $j = 1$ **to** $|\mathcal{B}|$ **do**
5:          Initialize a latent vector $\boldsymbol{\phi}_j^{(0)} \sim \mathcal{N}(\mathbf{0}, \sigma^2 \boldsymbol{I}) \in \mathbb{R}^H$.
6:          **for** $n = 0$ **to** $N_{\text{inner}} - 1$ **do**
7:              Update the latent vector: $\boldsymbol{\phi}_j^{(n+1)} \leftarrow \boldsymbol{\phi}_j^{(n)} - \epsilon_{\text{latent}} \nabla_{\boldsymbol{\phi}} \mathcal{L}(f_{\boldsymbol{\theta}, \boldsymbol{\phi}}(\mathbf{c}_j), \mathbf{y}_j)|_{\boldsymbol{\phi} = \boldsymbol{\phi}_j^{(n)}}$.
8:          **end for**
9:          Update the shared parameters: $\boldsymbol{\theta} \leftarrow \boldsymbol{\theta} - \epsilon_{\text{shared}} \nabla_{\boldsymbol{\theta}} \mathcal{L}(f_{\boldsymbol{\theta}, \boldsymbol{\phi}}(\mathbf{c}_j), \mathbf{y}_j)|_{\boldsymbol{\phi} = \boldsymbol{\phi}_j^{(N_{\text{inner}})}}$.
10:      **end for**
11: **end while**

---

**Algorithm 2** Auto-decoding with mNIF

1: Randomly initialize the shared parameter $\boldsymbol{\theta}$ of mNIF.
2: Initialize a latent vector $\boldsymbol{\phi}_j \sim \mathcal{N}(\mathbf{0}, \sigma^2 \boldsymbol{I}) \in \mathbb{R}^H$ for all samples.
3: **while** training **do**
4:      Sample a mini-batch $\mathcal{B} = \{(\mathbf{c}_j, \mathbf{y}_j)\}_{j=1:|\mathcal{B}|}$.
5:      Define a joint parameter: $\boldsymbol{\phi}_{\mathcal{B}} = \{\boldsymbol{\phi}_j\}_{j=1:|\mathcal{B}|}$
6:      Update the parameters: $\{\boldsymbol{\theta}, \boldsymbol{\phi}_{\mathcal{B}}\} \leftarrow \{\boldsymbol{\theta}, \boldsymbol{\phi}_{\mathcal{B}}\} - \epsilon \nabla_{\boldsymbol{\theta}, \boldsymbol{\phi}_{\mathcal{B}}} \sum_j \mathcal{L}(f_{\boldsymbol{\theta}, \boldsymbol{\phi}}(\mathbf{c}_j), \mathbf{y}_j)|_{\boldsymbol{\phi} = \boldsymbol{\phi}_j}$
7: **end while**

---

we introduce the sampling strategy for the context vectors $\boldsymbol{\phi}$ and customize it to a specific target task. To this end, we adopt the denoising diffusion probabilistic model (DDPM) [18], which employs the residual MLP architecture introduced in Functa [13].

## 5 Experiments

This section demonstrates the effectiveness of the proposed approach, referred to as mINF, and discusses the characteristics of our algorithm based on the results. We run all experiments on the Vessl environment [2], and describe the detailed experiment setup for each benchmark in the appendix.

### 5.1 Datasets and Evaluation Protocols

We adopt CelebA-HQ $64^2$ [21], ShapeNet $64^3$ [8] and SRN Cars [34] dataset for image, voxel and neural radiance field (NeRF) generation, respectively, where $64^2$ and $64^3$ denotes the resolution of samples in the dataset. We follow the protocol from Functa [13] for image and NeRF scene and generative manifold learning (GEM) [12] for voxel.

We adopt the following metrics for performance evaluation. To measure the reconstruction quality, we use mean-squared error (MSE), peak signal-to-noise ratio (PSNR), reconstruction Fréchet inception distance (rFID), reconstruction precision (rPrecision) and reconstruction recall (rRecall). In image generation, we use Fréchet inception distance (FID) score [17], precision, recall [32, 27] and F1 score between sampled images and images in a train split. Voxel generation performance is evaluated by coverage and maximum mean discrepancy (MMD) metrics [1] on a test split. In NeRF scene generation, we use FID score between rendered images and images in a test split for all predefined views for evaluation. To compare model size of algorithms, we count the number of parameters for training and inference separately; the parameters for training contain all the weights required in the training procedure, *e.g.*, the parameters in the mixture components and project matrix for our algorithm, while the model size for inference is determined by the parameters used for sampling instances. For the evaluation of efficiency, we measure the number of floating point operations per second (FLOPS), latency in terms of frames per second (fps), and the amount of memory consumption for a single sample.

Table 1: Image reconstruction and generation performance on CelebA-HQ $64^2$. The results of DPF, GEM, and Functa are brought from the corresponding papers.

| Model | # Params | | Reconstruction | | Generation | | | | Inference Efficiency | | |
|---|---|---|---|---|---|---|---|---|---|---|---|
| | Learnable | Inference | PSNR ↑ | rFID ↓ | FID ↓ | Precision ↑ | Recall ↑ | F1 ↑ | GFLOPS ↓ | fps ↑ | Memory (MB) ↓ |
| Functa [13] | 3.3 M | 2,629.6 K | 26.6 | 28.4 | 40.4 | 0.577 | 0.397 | 0.470 | 8.602 | 332.9 | 144.1 |
| GEM [12] | 99.0 M | 921.3 K | – | – | 30.4 | 0.642 | 0.502 | 0.563 | 3.299 | 559.6 | 70.3 |
| GASP [14] | 34.2 M | 83.1 K | – | – | 13.5 | 0.836 | 0.312 | 0.454 | 0.305 | 1949.3 | 16.4 |
| DPF [41] | 62.4 M | – | – | – | 13.2 | 0.866 | 0.347 | 0.495 | - | - | - |
| mNIF (S) | 4.6 M | 17.2 K | 31.5 | 10.9 | 21.0 | 0.787 | 0.324 | 0.459 | 0.069 | 2958.6 | 10.2 |
| mNIF (L) | 33.4 M | 83.3 K | 34.5 | 5.8 | 13.2 | 0.902 | 0.544 | 0.679 | 0.340 | 891.3 | 24.4 |

Table 2: Voxel reconstruction and generation performance on ShapeNet $64^3$. The results of DPF, and GEM are brought from the corresponding papers.

| Model | # Params | | Reconstruction | | Generation | | Inference Efficiency | | |
|---|---|---|---|---|---|---|---|---|---|
| | Learnable | Inference | MSE ↓ | PSNR ↑ | Coverage ↑ | MMD ↓ | GFLOPS ↓ | fps ↑ | Memory (MB) ↓ |
| GASP [14] | 34.2 M | 83.1 K | 0.0296 | 16.5 | 0.341 | 0.0021 | 8.7 | 180.9 | 763.1 |
| GEM [12] | 99.0 M | 921.3 K | 0.0153 | 21.3 | 0.409 | 0.0014 | 207.0 | 16.7 | 4010.0 |
| DPF [41] | 62.4 M | – | – | – | 0.419 | 0.0016 | - | - | - |
| mNIF (S) | 4.6 M | 17.2 K | 0.0161 | 20.9 | 0.430 | 0.0014 | 4.4 | 191.5 | 642.1 |
| mNIF (L) | 46.3 M | 83.3 K | 0.0166 | 21.4 | 0.437 | 0.0013 | 21.6 | 69.6 | 1513.3 |

## 5.2 Main Results

We present quantitative and qualitative results, and also analyze the effectiveness of the proposed approach in comparison to the previous works including Functa [13], GEM [12], GASP [14], and DPF [41]. Note that we adopt the model configurations with the best generation performance for the other baselines.

### 5.2.1 Quantitative Performance

We compare results from our approach, mNIF, with existing methods in terms of four aspects: reconstruction accuracy, generation quality, model size, and inference efficiency. We present quantitative results from two configurations of our model, mNIF (S) and mNIF (L), which correspond to small and large networks, respectively.

As shown in Tables 1, 2, and 3, our approach consistently achieves state-of-the-art reconstruction and generation quality on the image, voxel, and NeRF scene datasets except the reconstruction on the voxel dataset, in which our model is ranked second. Moreover, mNIF is significantly more efficient than other methods in terms of model size and inference speed in all cases. Considering the model size and inference speed, the overall performance of mNIF is outstanding.

Note that the light configuration of our model denoted as mNIF (S) is most efficient in all benchmarks and also outperforms the baselines in voxel and NeRF. In SRN Cars, mNIF (S) demonstrates huge benefit in terms of efficiency compared to Functa; 199 times less FLOPS, 49 times more fps, and 22 times less memory in single view inference. In the image domain, the performance of mNIF (S) is comparable to GASP in terms of F1 but worse in FID. We conjecture that FID metric overly penalizes blurry images, which is discussed in several works [12, 13, 31, 20].

### 5.2.2 Qualitative Generation Performance

Figure 2 illustrates generated samples for qualitative evaluation. We visualize the results from our models on each benchmark together with GASP on CelebA-HQ $64^2$ and Functa on SRN Cars to compare results in the image and NeRF scene tasks, respectively. In the image domain, our model, mNIF (L), generates perceptually consistent yet slightly blurry samples compared to ground-truths, while the examples from GASP with a similar FID to ours have more artifacts. In the NeRF scene, rendered views from our model, mNIF (S), and Functa exhibit a similar level of image blur compared to ground-truths. Since both methods use vanilla volumetric rendering, adopting advanced rendering techniques, such as hierarchical sampling [25, 6] and anti-aliasing methods [4, 3] for NeRF scenes, would improve rendering quality.

Table 3: NeRF scene reconstruction and generation performance on SRN Cars. The results of Functa are brought from the corresponding papers.

| Model | # Params | | Reconstruction | Generation | Inference Efficiency | | |
|---|---|---|---|---|---|---|---|
| | Learnable | Inference | PSNR ↑ | FID ↓ | TFLOPS ↓ | fps ↑ | Memory (GB) ↓ |
| Functa [13] | 3.9 M | 3,418.6 K | 24.2 | 80.3 | 1.789 | 2.0 | 28.0 |
| mNIF (S) | 4.6 M | 17.2 K | 25.9 | 79.5 | 0.009 | 97.7 | 1.3 |

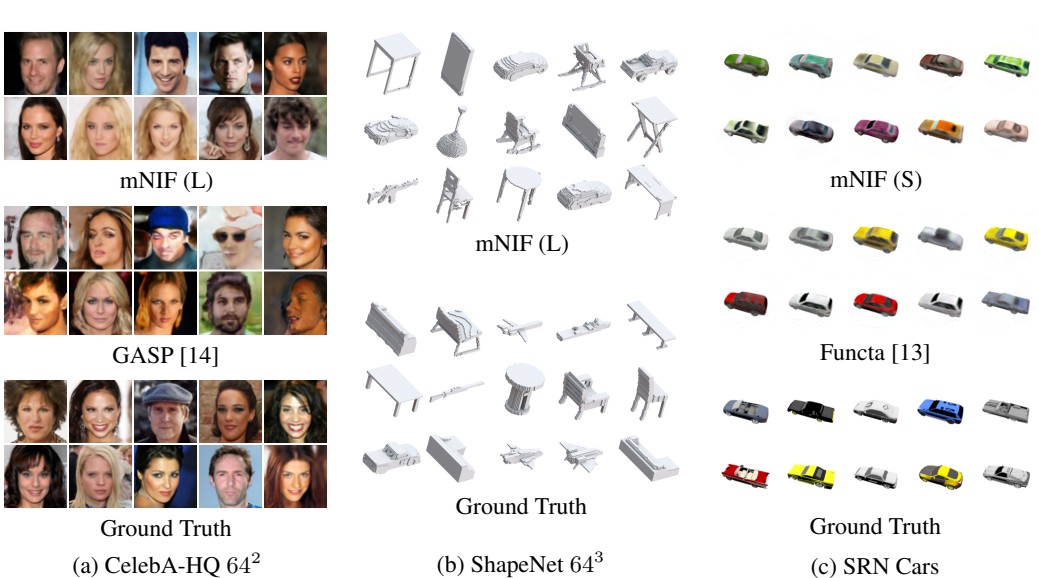

mNIF (L)

GASP [14]

Ground Truth

(a) CelebA-HQ $64^2$

mNIF (L)

Ground Truth

(b) ShapeNet $64^3$

mNIF (S)

Functa [13]

Ground Truth

(c) SRN Cars

Figure 2: Comparison of generated samples from our models, mNIF (S) and mNIF (L), with the ground-truth and the baseline methods such as GASP [14] and Functa [13], on the CelebA-HQ $64^2$ (2a), ShapeNet $64^3$ (2b) and SRN Cars (2c) datasets.

## 5.3 Analysis

We analyze the characteristics of our trained model for better understanding via latent space exploration and various ablation studies.

### 5.3.1 Latent Space Interpolation

Figure 3 illustrates the results of interpolation in the latent space; the corner images are samples in the training dataset and the rest of the images are constructed by bilinear interpolations in the latent space. These images demonstrate the smooth transition of sampled data, confirming that the latent vectors learned at the first stage manage to capture the realistic and smooth manifold of each dataset.

### 5.3.2 Configurations of Mixture Coefficients in mNIF

Table 4(a) examines the performance by varying the types of mixture coefficients: i) shared mixture coefficients across all layers, *i.e.*, $\alpha_m^{(i)} = \alpha_m$, ii) layer-specific mixture coefficients, and iii) layer-specific mixture coefficients projected from a latent vector as shown in Eq. (10). Among the three mixture settings, the last option yields the best performance and is used for our default setting.

Table 4(b) presents the effect of the number of mixtures, denoted by $M$. Reconstruction performance generally improves as the number of mixtures increases, but such a tendency saturates at around $M = 256$. We also measure the precision and recall between the ground-truth and reconstructed samples denoted by rPrecision and rRecall, respectively. According to our observation, increasing the size of the mixture improves the recall more than it does the precision.

Table 4(c) shows the effect of the latent dimension on the reconstruction performance of mNIF. Increasing the latent dimension $H$ leads to improving the reconstruction performance in both the train and test splits due to the degree-of-freedom issue.

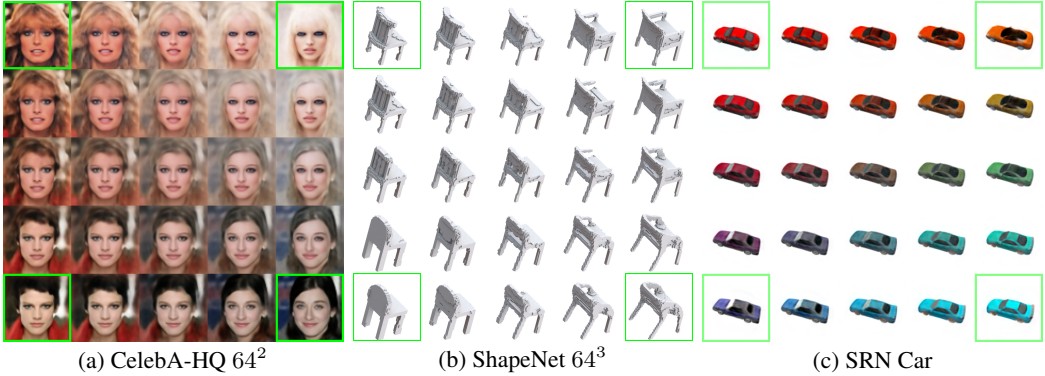

(a) CelebA-HQ $64^2$      (b) ShapeNet $64^3$      (c) SRN Car

Figure 3: Reconstructed images via latent vector interpolations. The examples at the four corners surrounded by green boxes are reconstructed instances using the latent vectors on CelebA-HQ $64^2$, ShapeNet $64^3$, and SRN Cars. The rest of samples are generated via bilinear interpolations of latent vectors corresponding to the images at the corners.

Table 4: Ablation study on the mixture coefficient configuration of mNIF on CelebA-HQ $64^2$. Note that $L$, $W$, $M$, and $H$ denote hidden layer depth, hidden layer width, the number of mixtures, and the dimensionality of a latent coefficient vector, respectively.

| Exp | Mixture | $(L, W, M, H)$ | # Params | | Train | | | | Test | |
|---|---|---|---|---|---|---|---|---|---|---|
| | | | Learnable | Inference | PSNR ↑ | rFID ↓ | rPrecision ↓ | rRecall ↓ | PSNR ↑ | rFID ↓ |
| (a) | Shared | $(2, 64, 64, 64)$ | 557.2 K | 8.7 K | 22.20 | 50.70 | 0.497 | 0.003 | 22.11 | 55.87 |
| | Layer-specific | $(2, 64, 64, 256)$ | 557.2 K | 8.7 K | 24.45 | 38.23 | 0.461 | 0.013 | 24.35 | 42.65 |
| | Latent | $(2, 64, 64, 256)$ | 623.0 K | 8.7 K | 25.27 | 31.67 | 0.534 | 0.040 | 25.09 | 36.85 |
| (b) | Latent | $(2, 64, 16, 256)$ | 155.8 K | | 22.02 | 53.01 | 0.433 | 0.001 | 21.84 | 57.23 |
| | | $(2, 64, 64, 256)$ | 623.0 K | 8.7 K | 25.27 | 31.67 | 0.534 | 0.040 | 25.09 | 36.85 |
| | | $(2, 64, 256, 256)$ | 2.5 M | | 26.64 | 24.62 | 0.640 | 0.134 | 25.74 | 32.17 |
| | | $(2, 64, 1024, 256)$ | 10.0 M | | 26.84 | 23.71 | 0.642 | 0.155 | 25.85 | 32.14 |
| (c) | Latent | $(5, 128, 256, 256)$ | 21.8 M | | 31.17 | 10.65 | 0.890 | 0.750 | 25.35 | 31.58 |
| | | $(5, 128, 256, 512)$ | 22.3 M | 83.3 K | 32.11 | 8.96 | 0.918 | 0.845 | 27.92 | 24.79 |
| | | $(5, 128, 256, 1024)$ | 23.2 M | | 32.71 | 8.09 | 0.935 | 0.893 | 29.45 | 22.05 |

### 5.3.3 Diversity in Learned Neural Bases

To investigate the diversity of the basis functions per layer, Figure 4 visualizes the absolute value of the pairwise cosine similarity between the weight matrices of the basis models. In the cosine similarity matrices, an element with a small value indicates that the corresponding two neural basis functions are nearly orthogonal to each other, which serves as evidence of diversity in the learned basis functions. The visualization also reveals the following two characteristics. First, the lower layers are more correlated than the upper ones partly because low-level features inherently have less diversity. Second, a small subset of neural bases have high correlations with others. We hypothesize that the concentration of high correlation among the small subset of neural bases could be mitigated by introducing a loss function enforcing orthogonality.

### 5.3.4 Context Adaptation Strategies

We perform the ablation study with respect to diverse optimization strategies and demonstrate the resulting reconstruction performance in Table 5. We train small mNIFs with $(L, W, M, H) = (2, 64, 256, 256)$ on the CelebA-HQ $64^2$ dataset. It is clear that the meta-learning approach using second-order gradient computation yields favorable results compared to other options. Running more inner-loop iterations marginally improves performance, but at the cost of additional computational complexity in time and space. Therefore, we set the number of inner-loop iterations to 3 for all meta-learning experiments.

Interestingly, the auto-decoding strategy surpasses meta-learning with first-order gradient computation. Note that auto-decoding is more efficient than other methods in terms of both speed and memory usage when evaluated under the same optimization environment. This efficiency of auto-decoding is

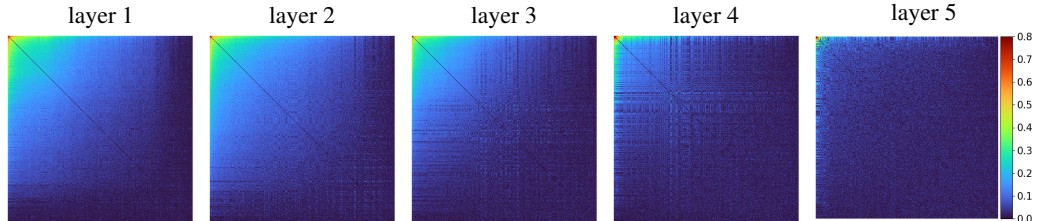

Figure 4: Visualization of the absolute value of cosine similarity between neural basis functions, observed in five different layers. We compute the similarity of the hidden layers in mNIF (L) trained on CelebA-HQ $64^2$. Diagonal values in these matrices are set to zero for visualization.

Table 5: Study on context adaptation strategy.

| Strategy | Second-order | $N_{\text{inner}}$ | PSNR ↑ | rFID ↓ |
|---|---|---|---|---|
| Auto-decoding | - | - | 24.68 | 34.31 |
| Meta-learning | ✗ | 3 | 22.68 | 57.27 |
| | ✓ | 3 | 26.64 | 24.64 |
| | ✓ | 5 | 26.77 | 24.37 |

Table 6: Study on longer training of mNIF.

| Epoch | Reconstruction | | Generation |
|---|---|---|---|
| | PSNR ↑ | rFID ↓ | FID ↓ |
| 400 | 32.11 | 8.96 | 14.96 |
| 800 | 33.10 | 7.31 | 14.30 |
| 1200 | 33.61 | 6.91 | 15.25 |

particularly beneficial for NeRF scene modeling, which requires a large number of input coordinate queries. Our model runs on a single GPU for training on 8 NeRF scenes per batch while Functa needs 8 GPUs for the same purpose.

### 5.3.5 Longer Training of Context Adaptation

Table 6 shows the performance of our approach, mNIF, with respect to the number of epochs for the first stage training under the identical stage 2 training environment. When we train mNIF (L) with $(L, W, M, H) = (5, 128, 256, 512)$ on the CelebA-HQ $64^2$ dataset, reconstruction performance is positively correlated with training length but longer training may be prone to overfitting considering the trend of generalization performance.

## 6 Conclusion

We presented a novel approach for generative neural fields based on a mixture of neural implicit functions (mNIF). The proposed modulation scheme allows us to easily increase the model capacity by learning more neural basis functions. Despite the additional basis functions, mNIF keeps the network size for inference compact by taking advantage of a simple weighted model averaging, leading to better optimization in terms of latency and memory footprint. The trained model can sample unseen data in a specific target task, where we adopt the denoising diffusion probabilistic model [18, 13] for sampling context vectors $\phi$. We have shown that our method achieves competitive reconstruction and generation performance on various domains including image, voxel data, and NeRF scene with superior inference speed.

The proposed method currently exhibits limited scalability beyond fine-grained datasets. We hypothesize that the latent space in mNIF trained on a dataset with diverse examples, such as CIFAR-10, is not sufficiently smooth compared to other datasets to be generalized by adopting a diffusion process. A possible solution to these challenges is to incorporate local information into the architecture based on neural implicit representations, as proposed in recent research [39, 5]. We reserve this problem for future work.

## Acknowledgments and Disclosure of Funding

This work was partly supported by the National Research Foundation of Korea (NRF) grant [No. 2022R1A2C3012210, No.2022R1A5A708390811] and the Institute of Information & communications Technology Planning & Evaluation (IITP) grant [NO.2021-0-0268, No. 2021-0-01343], funded by the Korea government (MSIT).

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

# A   Implementation Details

## A.1   Benchmark

### A.1.1   Images

We use CelebA-HQ dataset [21] for our work. We divide entire images into 27,000 images for train and 3,000 images for test split which is provided by Functa [13]. We use the pre-process dataset from the link [1]. To measure the quality of generated images, we compute Fréchet inception distance (FID) score [17], precision and recall [32, 27] between sampled images and images in a train split.

### A.1.2   Voxels

We utilize the ShapeNet dataset [8] for 3D voxel generation from IM-Net [10]. It has 35,019 samples in the train split and 8,762 samples in the test split. The dataset contains a voxel with $64^3$ and 16,384 points extracted near the surface. To evaluate the generative model, we sample 8,764 shapes, which is the size of the test set, and generate all points in the voxel. Then, we compute 2,048 dimension mesh features from the voxel following the protocol in [10]. Generation performance with coverage and maximum mean discrepancy (MMD) metrics [1] is evaluated using Chamfer distance on test split. We follow the evaluation procotol from generative manifold learning (GEM) [12].

### A.1.3   Neural Radiance Field (NeRF)

We adopt SRN Cars dataset [34] for NeRF modeling. SRN cars dataset has a train, validation, and test split. Train split has 2,458 scenes with $128 \times 128$ resolution images from 50 random views. Test split has 704 scenes with $128 \times 128$ images from 251 fixed views in the upper hemisphere.

We adopt the evaluation setting of NeRF scene from functa which utilizes 251 fixed views from the test split. Consequently, we can compute FID score based on rendered images and images in a test split with equivalent view statistics.

## A.2   Implicit Neural Representation with Mixture of Neural Implicit Functions

### A.2.1   Images

We train our mixtures of Neural Implicit Functions (mNIF) on images from train split of CelebA-HQ $64^2$ dataset. Generative implicit neural representation (INR) for image takes 2D input coordinates $(x_1, x_2)$ and returns 3D RGB values $(y_R, y_G, y_B)$. During optimization, we use dense sampling, which queries all input coordinates $(4096 = 64 \times 64)$ to the network inputs, for image benchmark. We use mNIF configuration with the number of hidden layers $L = 5$, the channel dimension of hidden layers $W = 128$, the number of mixtures $M = 384$ and the latent vector dimension $H = 512$. throughout our experiments. We use meta learning for training the mNIF on CelebA-HQ $64^2$. We use Adam [22] optimizer with the outer learning rate $\epsilon_{\text{outer}} = 1.0 \times 10^{-4}$ and batch size 32. We use a cosine annealing learning rate schedule without a warm-up learning rate. We take the best mNIF after optimizing the model over 800 epochs with inner learning rates $\epsilon_{\text{inner}} \in \{10.0, 1.0, 0.1\}$. The configuration of mNIF (S) is $(L, W, M, H) = (4, 64, 256, 1024)$ and mNIF (L) $(5, 128, 384, 512)$. The best performed model uses $\epsilon_{\text{inner}} = 1.0$.

### A.2.2   Voxels

We train our model on ShapeNet $64^3$ dataset. Our generative neural field for voxel takes 3D input coordinates $(x_1, x_2, x_3)$ and returns $y_\sigma$ indicating whether queried coordinate are inside or outside. For model optimization, we use sub-sampling with 4,096 points in a voxel with $64^3$ from the points. Then, we use all 16,384 points for constructing latent vectors. Note that despite relying on sub-sampling, the 16,384 points sampled from the near surface of a voxel are dense enough to effectively represent its content within the voxel. We use meta learning for training the mNIF on ShapeNet $64^3$. We use mNIF (S) and (L) configuration with $(L, W, M, H) = (4, 64, 256, 512)$ and $(L, W, M, H) = (5, 128, 512, 1024)$. We take the best mNIF after optimizing the model over 400 epochs with inner learning rates $\epsilon_{\text{inner}} \in \{10.0, 1.0, 0.1\}$.

---

[1] `https://drive.google.com/drive/folders/11VzOfqHS2rXDb5pprgTjpD7S2BAJhi1P`

### A.2.3 NeRF Scenes

We train our model on SRN Cars dataset. We use a simplified NeRF protocol without a view dependency on neural fields. So, generative neural field returns 4D outputs, RGB values and density $(y_R, y_G, y_B, y_\sigma)$, with given 3D input coordinates $(x_1, x_2, x_3)$. We use no activation function for RGB output and exponential linear unit (ELU) [11] for density output as suggested in Functa. During optimization, we use sparse sampling with 32 views per scene, 512 pixels per view and 32 points per ray for each scene. We use auto-decoding procedure for efficient optimization instead of meta-learning procedure to avoid the cost of second-order gradient computation. In auto-decoding procedure, we simply harvest latent vectors for the dataset jointly trained during the optimization procedure. We use mNIF (S) configuration with $(L, W, M, H) = (4, 64, 256, 128)$. We use Adam [22] optimizer with outer learning rate $\epsilon_{\text{outer}} = 1.0 \times 10^{-4}$ and batch size 8. We use cosine annealing without a warm-up learning rate for the learning rate schedule. We use inner learning rate $\epsilon_{\text{inner}} = 1.0 \times 10^{-4}$ and use weight decay on latent vector $\phi_j$ to regularize the latent space as in auto-decoding framework. We take the best mNIF after optimizing the model over 2000 epochs.

### A.2.4 Weight Initialization of mNIF

Weight initialization is crucial for optimizing INR, such as SIREN [33]. We enforce mixture coefficients at the start as the inverse of the number of mixtures $\alpha_m^{(i)} = 1/M$ to give the equivalent importance for all implicit bases in mixtures. For latent mixture coefficients, we set bias of projection matrix into $1/M$ to satisfy above condition.

### A.3 Denoising Diffusion Process on Latent Space

We train unconditional generation models for latent space for image, voxel and NeRF scene for generating latent coefficients vector. We use the equivalent setting for the training diffusion process throughout all experiments and benchmarks. We implement DDPM [18] on latent vector space based on residual MLP [13] from Functa as a denoising model. The channel dimension of residual MLP is 4096 and the number of block is 4. We use Adam optimizer and set learning rate $1.0 \times 10^{-4}$ and batch size 32. We train model 1000 epochs on each dataset and schedule the learning rate with cosine annealing without a warm-up learning rate. We use $T = 1000$ time steps for diffusion process.

### A.4 Libraries and Code Repository

Our implementation is based on PyTorch 1.9[2] with CUDA 11.3 and PyTorch Lightning 1.5.7.[3] Our major routines are based on the code repositories for Functa,[4] SIREN,[5] latent diffusion model,[6] guided diffusion[7] and HQ-Transformer.[8] For inference efficiency, we count FLOPS with the fvcore library[9] and check inference time and memory consumption on GPU by PyTorch Profiler.[10] The efficiency of mNIF are evaluated on NVIDIA Quadro RTX 8000.

## B Analysis

### B.1 Visualization

We provide additional generation examples in Figures 5 and 6 and additional interpolation results of the proposed algorithm in Figures 7 and 8.

---

[2]PyTorch (https://pytorch.org)

[3]PyTorch Lightning (https://www.pytorchlightning.ai)

[4]Functa (https://github.com/deepmind/functa)

[5]SIREN (https://github.com/vsitzmann/siren)

[6]Latent Diffusion Model (https://github.com/CompVis/latent-diffusion)

[7]Guided diffusion (https://github.com/openai/guided-diffusion)

[8]HQ-Transformer (https://github.com/kakaobrain/hqtransformer)

[9]fvcore (https://github.com/facebookresearch/fvcore/blob/main/docs/flop_count.md)

[10]PyTorch Profiler (https://pytorch.org/tutorials/recipes/recipes/profiler_recipe.html)

## C  Limitations

As discussed in concolusion, we observe that our model has limited scalability beyond fine-grain dataset; our model works well on reconstruction while not on generation on CIFAR10. To analyze the generation on CIFAR10, we introduce an alternative sampling scheme to replace the latent diffusion model described in our main paper. Drawing inspiration from the sampling strategy in generative manifold learning (GEM), we employed latent space interpolation between two samples: one random sample from the training set and another random sample from its neighborhood. The resulting context vector, is formed by a simple interpolation:

$$\phi_{\text{sample}} = \alpha \cdot \phi_i + (1 - \alpha) \cdot \phi_{\mathcal{N}(i)}, \tag{11}$$

where $\phi_i$ denotes context vector for $i$-th sample in training set, $\mathcal{N}(i)$ is a set of indices of the neighborhood of sample and is sampled from the uniform distribution ranged between 0 to 1.

This sampling by interpolation scheme achieves 59.67 (FID) while sampling from diffusion model does 84.64 (FID). From the results, two major observations can be made: Sampling through interpolation yields a superior FID score than the latent diffusion model, suggesting issues with the latter's interaction with context vectors. The FID score for interpolation sampling, while better than diffusion, remains suboptimal. The observation that the context vector interpolation is not fully effective for the CIFAR10 model indicates a less smooth latent space in CIFAR10 compared to other datasets. This disparity could be impacting the generation quality.

We hypothesize that the limitations of the SIREN [33] architecture might be at the root of these issues. An INR work [39] categorizes SIREN as an INR with predominantly frequency encodings and a lack of space resolution, leading to compromised content quality. A separate study [5] underlines this, stating SIREN's unsatisfactory generation performance (FID 78.2) when applied to CIFAR10. This research introduces a new SIREN architecture where biases in hidden layers of INR are varying with respect to spatial resolution. We anticipate that adopting an INR with spatial information as an alternative to SIREN will better accommodate diverse datasets, given that our approach—a linear combination of INR weights—is broadly compatible with various INR methods.

## D  Broader Impacts

The proposed image model is trained on the CelebA-HQ $64^2$ dataset, which raises the possibility of unintended bias in the generated human face images due to the biases present in the dataset. Additionally, when extending the generative neural field model to large-scale datasets, there is a potential for generating unintended biased contents, although the current version of the work primarily focuses on small-scale datasets. We acknowledge that these concerns are important themes that should be thoroughly discussed in the future.

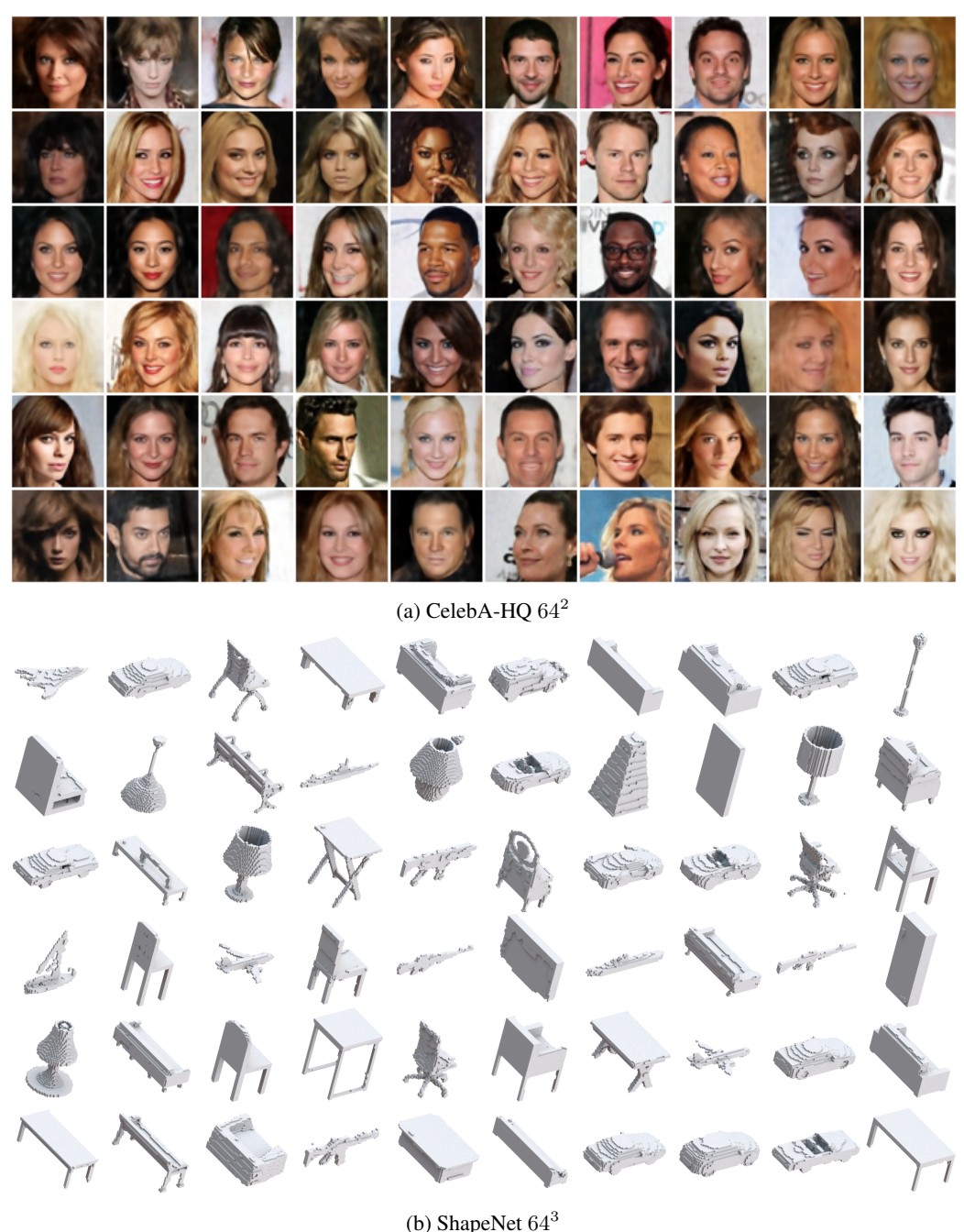

(a) CelebA-HQ $64^2$

(b) ShapeNet $64^3$

Figure 5: Generated samples from our models trained on CelebA-HQ $64^2$ (5a) and ShapeNet $64^3$ (5b).

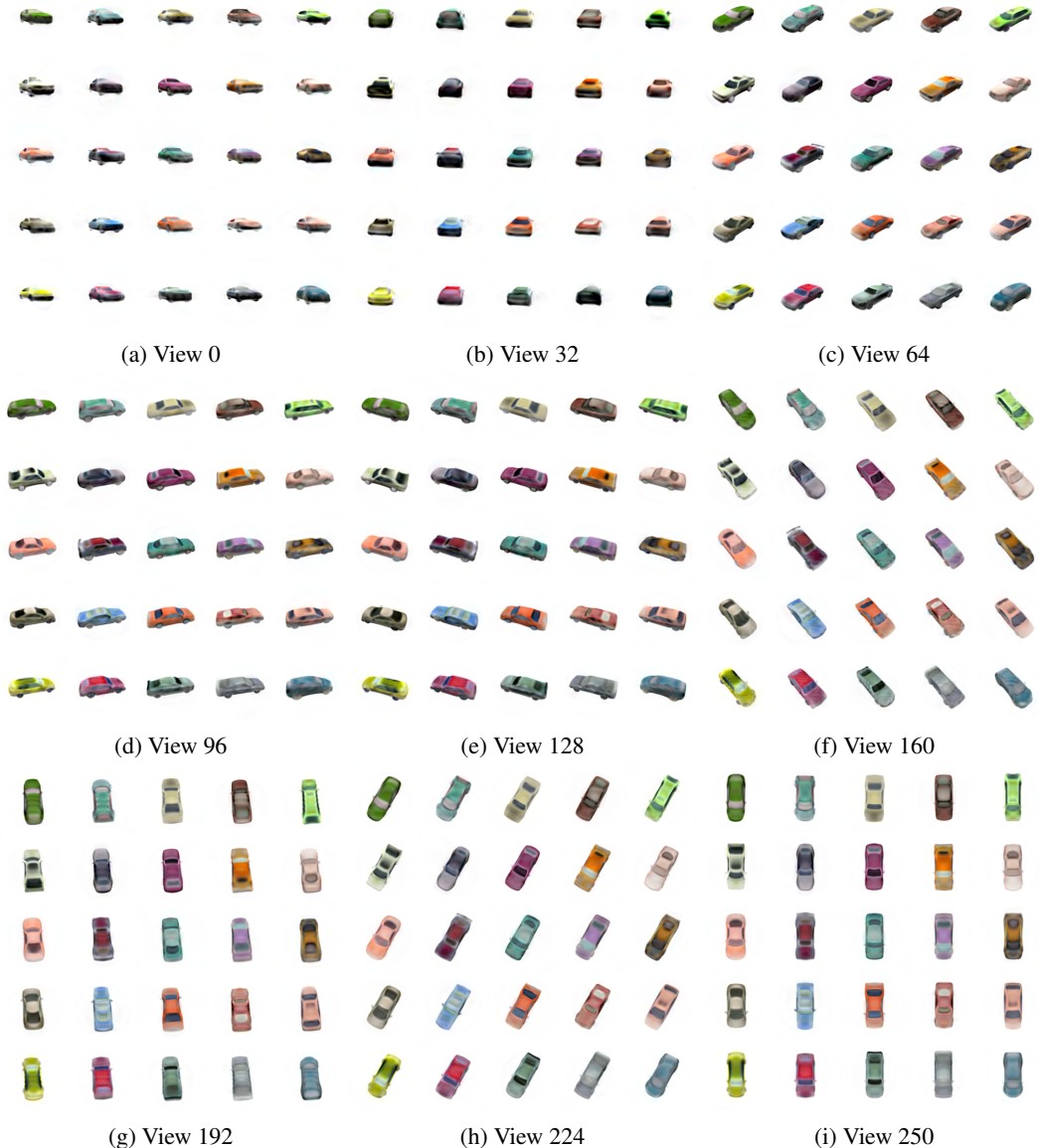

Figure 6: Generated 25 samples with diverse 9 views from our model trained on SRN Cars.

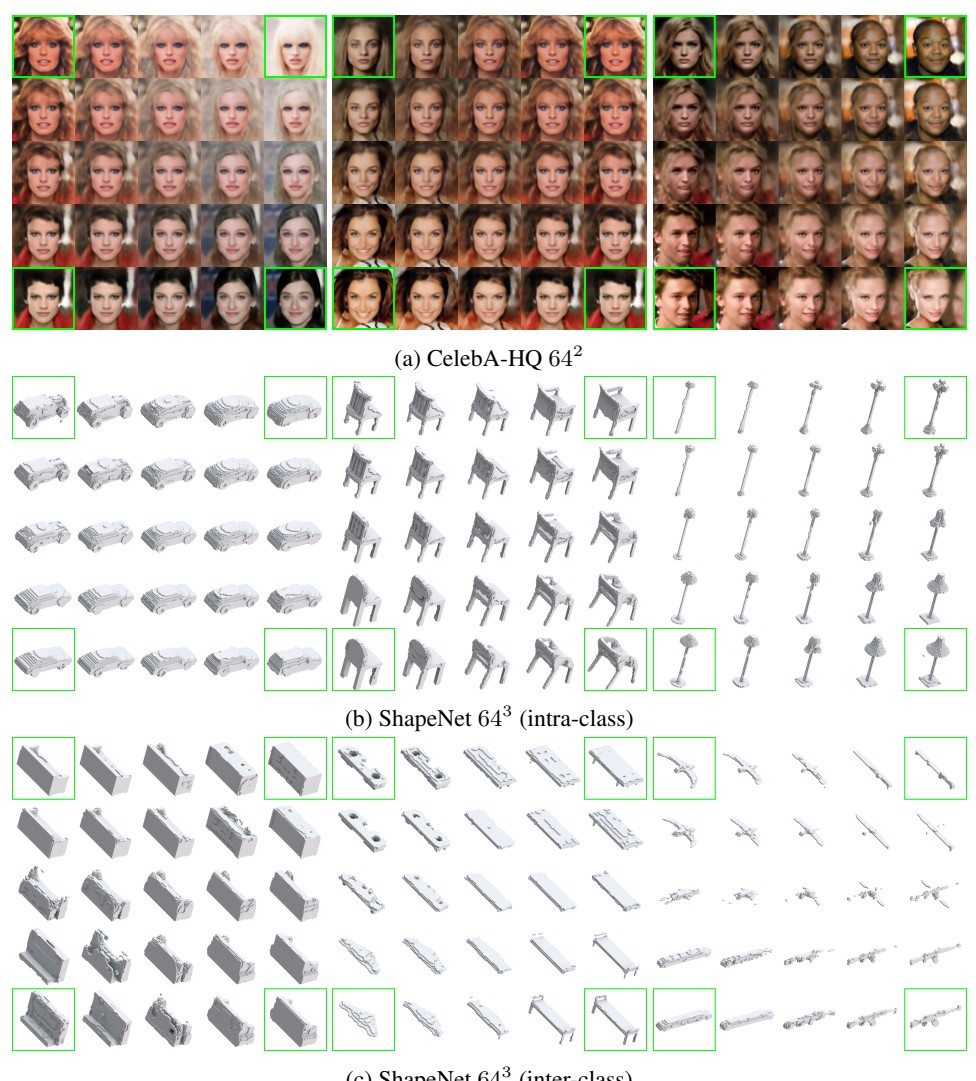

(a) CelebA-HQ $64^2$

(b) ShapeNet $64^3$ (intra-class)

(c) ShapeNet $64^3$ (inter-class)

Figure 7: Reconstructed samples from latent vectors for images (7a) and voxels (7b, 7c). Samples with green boxes are reconstructed instances from latent vectors estimated from train split for each dataset. The other samples are generated with bilinear interpolation of latent vectors from corner instances. In the third row 7c, we plot three interpolations with four different classes correspondingly. (left) chair (top left), cabinet (top right), display (bottom left), sofa (bottom right) (middle) loudspeaker, sofa, airplane, bench (right) airplane, bench, car, rifle. We use Mitsuba3 renderer for drawing voxel image.

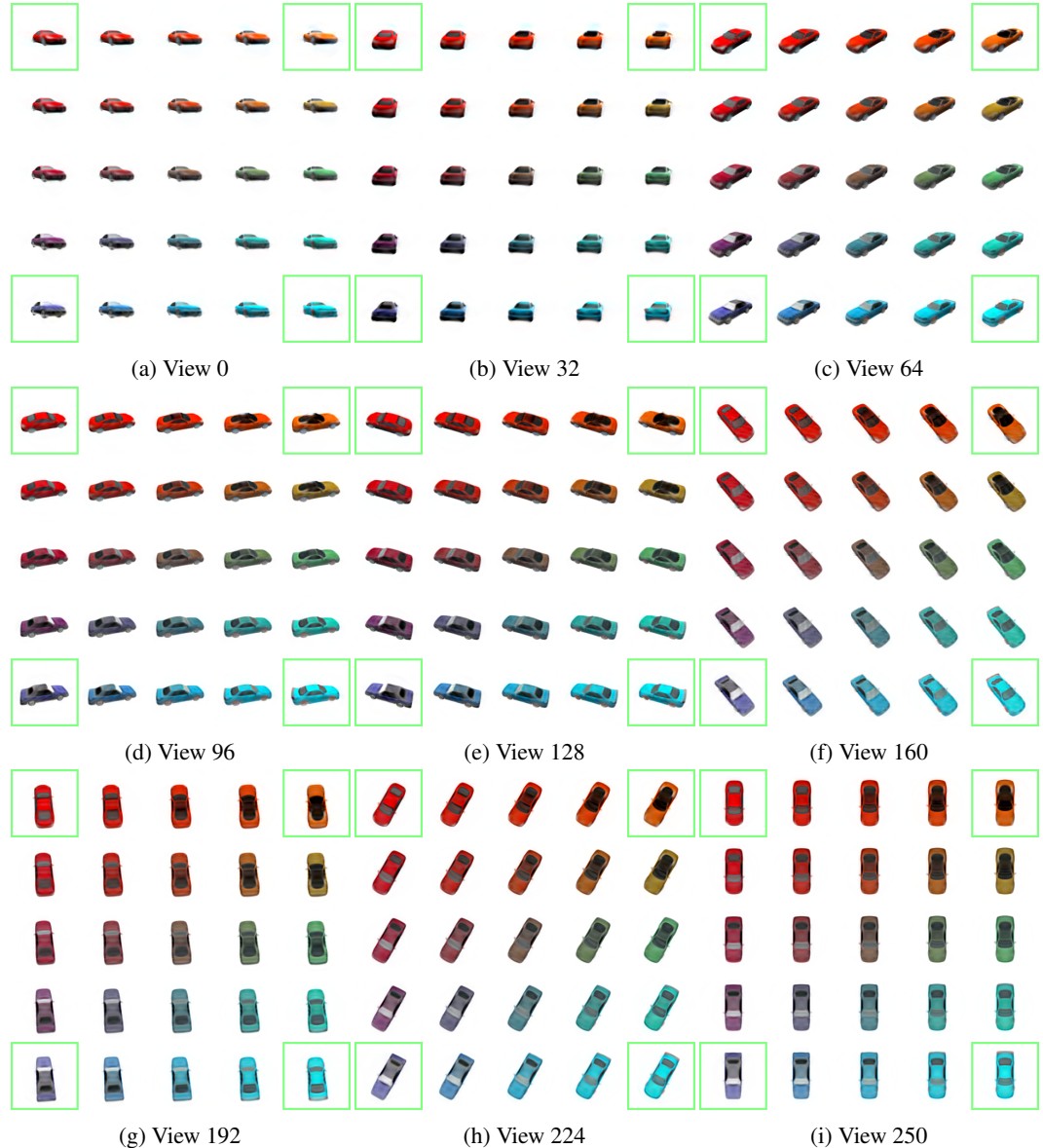

(a) View 0       (b) View 32       (c) View 64

(d) View 96       (e) View 128       (f) View 160

(g) View 192       (h) View 224       (i) View 250

Figure 8: Reconstructed view images from latent vector for NeRF scene. Image annotated by green boxes are synthesized instances from latent vectors estimated from SRN Cars train split. We select 9 views from the pre-defined 251 views in test split. The other samples are generated by bilinear interpolation of samples at the corners.

