# OpenReview forum: "Generative Neural Fields by Mixtures of Neural Implicit Functions"
_NeurIPS.cc/2023/Conference — NeurIPS 2023 poster_

### Official Review · Reviewer_YZcJ · 2023-07-07

**Soundness:** 2 fair
**Presentation:** 2 fair
**Contribution:** 3 good
**Rating:** 5
**Confidence:** 4

**Summary:**

This paper proposes a new representation of neural fields based on a mixture of neural fields. Neural fields is another name for implicit neural representation (INR) which is a representation of data points (like images or 3d scenes), that maps an input of 2d or 3d position into the output of the RGB color and/or shape information.

The paper builds on previous work like GASP [12], Functa [13] and GEM [1] and proposes a new representation on which the generative model is trained. As in [13] the generative model is a diffusion model and the novelty here is the representation which oi based on a mixture model.The mixture model representation proposed is implemented as a shared set of INRs which weights are averaged with a vector of mixture weights that are optimized per data-point.

After describing the model, the paper demonstrates its effectiveness in different data modalities like images (Celeba-HQ), 3D shapes (ShapeNet) and 3D scenes (SRN Cars), where the model performs well compared to previous approaches.  Then some additional ablation and analysis are presented.

**Strengths:**

The main strength of this paper is the novel INR representation proposed. This representation follows a recent line of work on the topic, that aims to leverage the advantages of INRs into the setup of training generative models on different data modalities. A main challenge there is capturing 3D data in an efficient way, which is still not acheivable in standard direct methods.
The proposed model is an interesting direction with a potential to contribute to the research of this topic.

**Weaknesses:**

The main weaknesses of this paper are the following:
1. The description of the model and the comparisons to previous work is lacking. Although some of the aspects are simple and clearly presented and the comparison to previous models is extensive and contains many different metrics, there still remains some important aspects that are not sufficiently described (see questions below).
2. After describing the model, the authors "jump" to final metrics like PSNR and FIDs. Given that the main contribution of this work is the mixture representation itself, it is essential to show what this representation captures. Figure 5 is a good start in this direction, but what is still missing is some understanding of what the different components in the mixture are doing, and how they are combined. The authors could, for example, show the reconstructed images of different components in isolation as the number of components grow. Another component which is directly used in the model without investigation is the hierarchical mixture (mixture of every layer in the network). How does this affect the components? How are the differences in lower layer components manifested compared to differences in higher level components?

**Questions:**

1. The meta learning algorithm presented in algorithm 1:  why is $\phi$ randomaly initialized at every iteration? This is different than MAML that learns an initialization, and CAVIA [17] (which the authors state as reference for this) that keeps the initialization constant. This is an interesting difference that should be discussed.
2. Algorithm 2: The pseudo code is not clear about the indexes of $\phi$ how are they being updated through the iterations?
3. One aspect that is missing from all tables and comparisons is the size of the representation. For the proposed model this would be the number of mixture components times the number of layers. This is an important aspect of the model since it defines the rate-distortion tradeoff. The way the authors count parameters (learnable vs. inference) is interesting and of significant value, but adding the size of the representation is essential.
4. How are the previous models chosen for comparison? These models usually contain several options with different sizes of the network and the representation, where the reconstruction quality usually grows with the size of the network.
5. Line 115: What exactly is invariant to what?
6. The ablation in the first section of 5.3 is not very useful. As stated above, it would be better to try to understand  what every component in the mixture captures.
6. In figure 5, I'm not sure I understand what the plots show. Are the matrices sorted somehow?

**Limitations:**

The main limitation that is missing from the discussion in the paper is the relation of the quality of representation and its size (see weakness #2 and question #3).

---

> ### Author Rebuttal · Authors · 2023-08-10
>
> Thank you for complimenting our modulation scheme, mixtures of neural implicit functions for neural fields. Following part, we will describe the properties of our model to solve the raised concerns by the reviewer.
>
>
>
> ## W1 Comparison to previous work
>
> In our study, we focus on algorithms for generating neural fields; especially the algorithms that can handle **every modality such as 2D image, 3D voxel, and NeRF**. That’s why we take GEM, GASP, Functa as the baseline methods for our work, and we describe the comparison to the previous works on question 3 and 4 in below.
>
> ## W2 lack of analysis on the proposing method
>
>
> ### Ablations on mixture type, such as mentioned as hierarchical mixture
> In Section C.2 of our supplementary material, we investigated the reason for selecting current architecture among three types of mixture coefficient: shared, layerwise (referred to as hierarchical mixture by the reviewer), and latent.
> Building upon the observation, we adopt the latent mixture coefficient for all our models in the experiments. We will reference this ablation study in the main paper to ensure clarity.
>
> ### Deeper analysis on mixtures
> We have observed that the information for each sample is distributed quite uniformly across all mixtures within the hidden layers.
> In Figure A (a), we illustrate the images reconstructed by incrementally increasing the number of mixtures in each layer.
> To visualize this figure, we need an order for the neural implicit functions (NIF) within each mixture. We begin by projecting a context vector into layerwise coefficients, as our model leverages latent mixture coefficients to merge these coefficients. Subsequently, we calculate the mean values of the layerwise coefficients across all training samples and sort for computing significance of each NIF in mixtures for every layer.
> In Figure A (a), $M=0$ displays the reconstructed image using the mean vector of layerwise coefficients. $M=48$ presents the image reconstructed with layerwise coefficients from the 1st to the 48th important coefficients, with the remaining coefficients set to zero.
> Figure A (a) displays a gradual transformation of the reconstructed image. We hypothesize that this reflects the nature of the distributed representation in our methodology. This observation aligns with the visualization of diversity presented in Figure 5 of the main paper. This suggests that numerous NIFs contribute to the representation of an image.
>
>
> Figure A (b) shows reconstructed images relative to layer depth. In the top row, we illustrate the reconstructed image by progressively adding layerwise coefficients for the higher layers to the mean layerwise coefficients. In contrast, the bottom row demonstrates the inverse by systematically omitting layerwise coefficients from the lower to the higher layers. Notably, both the addition of the final layer and the removal of the initial layer have minimal impact on the reconstruction quality. Unlike the input and output layers, the transitions among the hidden layers are more pronounced. From these observations, we infer that the mixtures within all hidden layers encapsulate overarching information from the datasets. In contrast, the mixtures in the input and output layers seem to capture comparatively less information than the hidden layers.
>
> ## Q1 initialization of context vector
> We observe that random initialization is more stable than constant initialization in CAVIA with a large number of mixtures ($M >= 256$) in our model.
> If the training process with CAVIA is done without collapsing, however, random initialization and constant initialization of context vectors.
>
> ## Q2 how to update $\phi$ in algorithm 2
> For the auto-decoding with mNIF, $\phi$ are parameters shared along all samples and $\psi_j$ is a latent vector of $j$-th sample, where both of them are learnable parameters.
> At every iteration, the shared parameter $\phi$ and the latent vectors of samples in the current batch $\psi_B$ are updated.
>
> ## Q3 dimension of context vector
> We use latent mixture coefficients with fixed hidden representation size denoted $H$. This hidden representation size is the dimension of context vector, because the projection matrix reduces the dimension of all alphas into the projected dimension as written in Eq 10.
>
> We use 512 in image projected from 7 $\times$ 384. and 1024 in voxel projected from 7 $\times$ 384. In 128 in NeRF projected from 6 \times 128.
> We set the size of context vector as the best performed configuration from each model. Functa uses 256 for image, 64 for NeRF. GEM uses 1024 for image and voxel. GASP uses 64 for image and voxel.
>
> ## Q4 How are the previous models chosen for comparison?
> We adopt the model configurations with **the best generation performance** from the previous works. While the configuration of the existing model was fixed, the configuration of the proposed model was changed.
>
> ## Q5 initialization of SIREN
> Invariant to output distribution at initialization.
>
> ## Q6 analysis on mixtures
> Please check the response to W2 in the above paragraph.
>
> ## Q7 how to visualize diversity in Figure 5
> To show the diversity of learned neural implicit functions, we compute the absolute value of cosine distance between two neural implicit functions in mixtures. This value becomes zero when two NIFs are orthogonal and becomes one if they are correlated.
>
> The initial visualization of these values appears as random noise, due to the absence of a pre-defined order for the NIFs in the mixtures. To order each mixture, we follow the procedure outlined below:
> First, we compute this value for all mixtures and acquire pairwise value matrices for all mixtures.
> Second, we take the largest value from pairwise values and take the pair indices (i, j) for each mixture.
> Third, we add pair indices i and j into the ordered list if an index is not in the list.
> Fourth, iterate the second and the third step until consuming all indices for each mixture.

---

> > ### Author Response · Authors · 2023-08-15
> >
> > We strive to provide concise and clear responses, but we understand if our previous reply may not have fully addressed your concerns or could have been misunderstood. Please feel free to ask any further questions.
> >
> > **Additional Comment on Q1**
> > To elaborate on what might have been missing in our prior response regarding the random initialization of \phi: As previously discussed, during meta-training, our model begins with a random latent vector sampled from a zero-mean Gaussian distribution with a standard deviation of \sigma. This is a slightly different approach taken by CAVIA. Our intention in doing so is to diversify INR weights at the start phase of meta-training, ensuring that not all samples are represented with identical INR weights.

---

> > > ### Comment · Reviewer_YZcJ · 2023-08-17
> > >
> > > Thank you for your answer. Regarding Q3 and the dimension of representation, I think this means you have a typo in line 139. Shouldn't the vector of $\alpha$'s be of dimension $M \times (L+2)$ rather than $\phi$?
> > >
> > > I still think the representation size is an important aspect when comparing the different methods, and should be part of the comparison tables.

---

> > > > ### Author Response · Authors · 2023-08-17
> > > >
> > > > We made a mistake that led you to misunderstand the content. Line 139 does have a typo; the correct notation should be $\phi \in \mathcal{R}^{H}$. The dimension $M \times (L+2)$ represents the total number of $\\{ \alpha_m^{(l)} \\}$.
> > > >
> > > > Our model used latent mixture coefficients, and the latent coefficient vector $\phi$ serves as the representation for a sample. So, the size of the coefficient vector $H$ becomes the representation size.  As mentioned in answer to Q3, we set the representation size of our model as comparable to other models in the experiment tables of the main text.
> > > >
> > > > We are sorry for the mistakes and we will correct this error and explicitly mention the representation size in the tables, so it can be compared with other methods.
> > > >
> > > > We appreciate your careful and thorough review. Please don't hesitate to reach out if you have any further questions or concerns.

---

> > > > > ### Author Response · Authors · 2023-08-21
> > > > >
> > > > > **Updating Table with representation size**
> > > > >
> > > > > To clarify your request, we instantiate the extended version of Table 1 of the main paper with the additional representation size denoted as $D(\phi)$.
> > > > >
> > > > > | Model | # Learnable Param | # Inference Param | $D(\phi)$ | PSNR | rFID | FID | Precision | Recall | F1 |
> > > > > |----------|-----------------|-----------------|-------------|---------|-------|-------|---------|----------|---------|
> > > > > Functa    | 3.3 M           | 2,629.6 K     | 256          | 26.6 | 28.4 | 40.4 | 0.577 | 0.397 | 0.470 |
> > > > > Functa    | 4.5 M           | 2,629.6 K     | 512          | 29.7 | 17.2 | -        | -          | -         | -          |
> > > > > Functa    | 6.8 M           | 2,629.6 K     | 1024        | 32.4 | -       | -        | -          | -         | -          |
> > > > > GEM       | 99.0 M         | 921.3 K        | 1024        | -        | -       | 30.4 | 0.642 | 0.502 | 0.563 |
> > > > > GASP      | 34.2 M         | 83.1 K           | 64            | -        | -       | 13.5 | 0.836 | 0.312 | 0.454 |
> > > > > DPF         | 62.4 M         | -                    | -                | -        | -       | 13.2 | 0.866 | 0.347 | 0.495 |
> > > > > mNIF (main paper) | 33.4 M | 83.3 K  | 512        | 34.5 | 9.2    | 13.2 | 0.902 | 0.544 | 0.679 |
> > > > >
> > > > > The performances of Functa, GEM, GASP, and DPF are adopted from the reported values in each paper.

---

> > > > > > ### Comment · Reviewer_YZcJ · 2023-08-22
> > > > > >
> > > > > > Thank you for your answers. This discussion removes some of the concerns I had regarding fair comparisons and lack of analysis, so I slightly raise my score.

---

> > > > > > > ### Author Response · Authors · 2023-08-22
> > > > > > >
> > > > > > > Thank you for your feedback, which has highlighted the need for comparison and deeper analysis to enhance understanding of the model. We also appreciate your improved rating of our work. We will revise our paper to include discussions over the periods.

---

### Official Review · Reviewer_ZMK5 · 2023-07-07

**Soundness:** 3 good
**Presentation:** 2 fair
**Contribution:** 2 fair
**Rating:** 5
**Confidence:** 5

**Summary:**

This paper deals with the problem of learning the latent space of data through the concept of meta-learning INRs (or neural fields). In contrast to prior works that consider either the adaptation of whole neural field parameters or some linear modulation parameters, the proposed method considers some basis neural field parameters as the shared parameters and adapts to coefficients for this basis to be adapted. The proposed method better encodes the given data of diverse modality, compared with prior INR-based data encoding scheme.

**Strengths:**

- The observation that the linear combination of neural fields as basis function parameters is quite interesting and novel.
- The quantitative results are better than prior works.
- The paper is generally well-written and easy to follow, while some important details are missing in the current status (see the weakness below)


**Weaknesses:**

- For generating new data from the proposed framework, specification of the latent sampling strategy (e.g., what is a prior distribution of $\phi$) is crucial, while the paper only states "off-the-shelf sampling method" in L179. Explaining how the sampling is performed would make the paper much stronger.
- I guess this method requires much more GPU memory, as it uses multiple basis neural field parameters with the (second-order) gradient-based meta-learning method. In particular, this method requires about 10 times more parameters than Functa.
- The experiments are mostly conducted under fine-grained datasets; I wonder if the method can be worked well with more complex datasets (e.g., CIFAR-10 or ImageNet-64x64) compared with other methods.


**Questions:**

- Is there any constraints (or ranges) for the coefficients $\alpha_i$s?

**Limitations:**

The paper addresses the limitations and negative social impacts in the supplementary materials.

---

> ### Author Rebuttal · Authors · 2023-08-10
>
> Thank you for complimenting the proposed model, experiments and writing quality. Our model can achieve both flexibility on model capacity and inference efficiency based on linear combination of model weights of neural fields, which is related to question 2. We answered your questions as below. We will revise our paper by reflecting your comments.
>
> ## W1. explanation on latent diffusion
> We adopt denoising diffusion probabilistic process (DDPM) [1] framework for modeling latent space for context vector. DDPM progressively denoises a random vector into a context vector during sampling from latent space. This sample process is done by the reverse process of DDPM. It is formulated as
>
> $p( \mathbf{x} _T ) = \mathcal{N} ( \mathbf{x} _T ; \mathbf{0} , \mathbf{I}), p _{\theta} ( \mathbf{x} _{0:T}) = p (\mathbf{x} _T \prod _{t=1}^{T} p _\theta (\mathbf{x} _{t-1} | \mathbf{x} _t ), p _\theta (\mathbf{x} _{t-1} | \mathbf{x} _t ) = \mathcal{N} (\mathbf{x} _{t-1} ; \mathbf{\mu} _\theta (\mathbf{x} _t , t), \mathbf{\Sigma} _\theta (\mathbf{x} _t , t))
> $
>
> We train a denoising model to denoise data by using the relationship between data and its corrupted one with Gaussian noise. This corrupting process is called forward process and formulated as follow.
>
> $q (\mathbf{x} _t | \mathbf{x} _0 ) = \mathcal{N} ( \mathbf{x} _t ; \sqrt{\overline{\alpha} _t } \mathbf{x} _0 , (1-\overline{\alpha} _t)\mathbf{I}, q(\mathbf{x} _t | \mathbf{x} _{t-1} ) = \mathcal{N} (\mathbf{x} _{t-1} ; \sqrt{1-\beta _t} \mathbf{x} _{t-1} , \beta _t \mathbf{I})$.
>
> where $\alpha _t = 1 -\beta _t$ and $\overline{\alpha} _t = \prod _{s=1}^{t} \alpha _s$.
>
> Mean and variance of the reverse conditionals $\mathbf{\mu} _\theta (\mathbf{x} _t , t), \mathbf{\Sigma} _\theta (\mathbf{x} _t , t)$ by neural network. We adopt residual MLP introduced in Functa for our denoising model in latent space.
> The parameter of denoising model can be optimized by minimizing
>
> $\mathbb{E} _q [ KL [ q(\mathbf{x} _{t-1} | \mathbf{x} _t , \mathbf{x} _0 || p _\theta (\mathbf{x} _{t-1} | \mathbf{x} _t ]]$
>
> We expect that tightly coupling two steps training procedure. A set of context vectors in our model is acquired by context adaptation procedure by Algorithm 1 or 2 in our paper. It is intialized as random vector and resulted into represent a single sample after the procedure. This process can be considered as a forward process in our model substituting fixed forward process by Gaussian noise.
>
>
>
> [1] Denoising Diffusion Probabilistic Models, NeurIPS 2020
>
> ## W2. comparison to Functa in terms of memory
>
> Our model consumes much less GPU memory than Functa during **both training and inference time**.
>
>
> The below table exhibits the GPU memory consumptions of our model and Functa; both models are trained with the meta-learning procedure on CelebA-HQ $64^2$ with batch size 32. Our method requires 4.3 and 5.9 times less GPU memory for training and inference, respectively, although it is true that our method requires more learnable parameters (Ours: 33.4M, Functa: 3.3M).
>
>
> Memory    | Ours		| Functa
> | ------------- | ------------- | ------ |
> Training (batch 32)	| 9.3 GB 	| 40.2 GB
> Inference 		| 24.4 MB	| 144.1 MB
>
>
> It is because that memory requirement for generative neural field is highly dependent on the size of inference parameter than that of learnable parameter due to the large memory consumption on tremendous input coordinate queries. This property is our main motivation for efficient design; aggregation INR weights within weight space. Note that our inference parameter is much smaller than those of Functa.
>
> As the reviewer mentioned, a second-order gradient method requires more memory if the learnable parameter becomes larger. However, due to the **tremendous input coordinate queries** of this generative neural field, the impact of inference parameters exceeds the impact of the learnable parameters. This reasoning can be validated by comparing memory consumption during training and inference. Although training requires more memory than inference due to the second-order gradient method, the gap between the two methods is still maintained. So, this observation reinforces our motivation for the proposed method; reducing inference parameters.
>
>
>
> ## W3. Beyond fine-grain dataset
> Training on a more complex dataset, such as CIFAR10, is a good suggestion, but unfortunately, we need more time to analyze a model on a new dataset. Currently, we observe that our model works well on reconstruction while not on generation. We will discuss this experiment during the author-discussion period.
>
>
> ## Q1. constraints for the coefficient
> We have no constraint on the coefficient except for the initialization. As explained in section B in the supplementary material, we set alphas as 1/M at initialization to satisfy SIREN initialization scheme for shared and layerwise mixture coefficients, where M is the number of mixtures.

---

> > ### Comment · Reviewer_ZMK5 · 2023-08-11
> > **Response**
> >
> > Thanks for the response. The provided response addressed most of my concerns. Please add the response of W2 in the final version of the manuscript if the paper is accepted. I slightly increase my score to 5, as the proposed method currently seems limited scalability to more complex dataset.

---

> > > ### Author Response · Authors · 2023-08-12
> > >
> > > Thank you for your insightful feedback and for the improved rating on our work. We will address the issues previously mentioned. If you have any questions regarding our work, please feel free to leave comments.

---

> > > > ### Author Response · Authors · 2023-08-20
> > > >
> > > > To analyze the generation on CIFAR10, we introduce an alternative sampling scheme to replace the latent diffusion model described in our main paper. Drawing inspiration from the sampling strategy in generative manifold learning (GEM) [2], we employed latent space interpolation between two samples: one random sample from the training set and another random sample from its neighborhood. The resulting context vector, $\phi_\text{sample}$, is formed by a simple interpolation:
> > > >
> > > > $\phi_\text{sample} = \alpha \cdot \phi_i + (1- \alpha ) \cdot \phi_{\mathcal{N}(i)}$
> > > >
> > > > where $\phi_i$ denotes context vector for $i$-th sample in training set, $\mathcal{N}(i)$ is a set of indices of the neighborhood of sample $i$ and $\alpha$ is sampled from the uniform distribution ranged between 0 to 1.
> > > >
> > > > method  | reconstruction (PSNR) | sampling by interpolation (FID)  | sampling from diffusion model (FID)  |
> > > > |----------|-------------------------------|-------------------------------------------|---------------------|
> > > > ours        | 29.89                               | 59.67                                               | 84.64                 |
> > > >
> > > > From the results, two major observations can be made:
> > > > - Sampling through interpolation yields a superior FID score than the latent diffusion model, suggesting issues with the latter's interaction with context vectors.
> > > > - The FID score for interpolation sampling, while better than diffusion, remains suboptimal.
> > > >
> > > > The observation that the context vector interpolation is not fully effective for the CIFAR10 model indicates a less smooth latent space in CIFAR10 compared to other datasets. This disparity could be impacting the generation quality.
> > > >
> > > > We hypothesize that the limitations of the SIREN [1] architecture might be at the root of these issues. Research [3] categorizes SIREN as an INR with predominantly frequency encodings and a lack of space resolution, leading to compromised content quality. A separate unpublished study [4] underlines this, stating SIREN's unsatisfactory generation performance (FID 78.2) when applied to CIFAR10. This research introduces a new SIREN architecture where biases in hidden layers of INR are varying with respect to spatial resolution. We anticipate that adopting an INR with spatial information as an alternative to SIREN will better accommodate diverse datasets, given that our approach—a linear combination of INR weights—is broadly compatible with various INR methods.
> > > >
> > > >
> > > > References
> > > > - [1] Implicit Neural Representations with Periodic Activation Functions, NeurIPS 19
> > > > - [2] Learning Signal-Agnostic Manifolds of Neural Fields, NeurIPS 21
> > > > - [3] Neural Fourier Filter Bank, CVPR 23
> > > > - [4] Spatial Functa: Scaling Functa to ImageNet Classification and Generation, ArXiv 2302.03130, February 6th 2023.

---

### Official Review · Reviewer_2vix · 2023-07-08

**Soundness:** 4 excellent
**Presentation:** 3 good
**Contribution:** 4 excellent
**Rating:** 7
**Confidence:** 3

**Summary:**

This paper proposes an approach to increase the learning capacity of implicit neural networks (INRs). The idea is to define a set of basis implicit functions that can be mixed to produce an instance-specific INR for each instance in a dataset. Notably, the approach does not affect the inference performance as once an instance-specific INR is produced, the cost of coordinate sampling is the same as with any other INR. This method allows implicit neural networks to be generalized on entire datasets rather than overfitting to a single instance while still demonstrating high fidelity of samples. The training procedure is split into two steps, the first step being a meta learning or auto decoding problem to fit the basis INRs with a reconstruction loss to the training data, and the second step is training a latent model to generate samples that determine the mixing coefficients. The paper demonstrates solid results and evaluation on different datasets of varying modalities including images, voxels and NeRFs.

**Strengths:**

The paper is well-written and easy to follow. The methodology is well explained and is simple but effective. The inference performance is significantly better than three state-of-the-art methods considered, and so are the extensive generation metrics like FID, precision, recall, coverage etc. The methodology and design choices are comprehensively evaluated on multiple datasets, and modalities with ablation studies.

**Weaknesses:**

The paper makes a solid contribution and I do not see any major flaws.

**Questions:**

Typos:
Line 108: resepctively -> respectively
Line 177: requires -> require

I was wondering if the authors could add more interpolation results as in Figure 4, perhaps in the supplementary material. In particular, it would be interesting to see how interpolation works across ShapeNet categories.

**Limitations:**

Yes, limitations and societal impact are sufficiently discussed in the supplementary material.

---

> ### Author Rebuttal · Authors · 2023-08-10
>
> Thank you for emphasizing the strengths of the proposed method and complimenting the current version of the work.
>
> ## Q1. additional interpolation plots
> We visualize additional inter-class interpolation results in Figure B of attached one-page PDF. We conduct bilinear interpolation with samples from four distinct classes in ShapeNet $64^3$.
> We can observe that inter-class interpolation is still working as in intra-class interpolation although there exist some clutters with interpolation of dissimilar objects. We will update those visualizations in the future version of the submission.

---

> > ### Author Response · Authors · 2023-08-15
> >
> > We truly appreciate your positive feedback on our work. We leave a comment to address the missing class information on the interpolation plots in Figure B of the attached PDF:
> > For the objects in positions (a): they are chair, cabinet, display, and sofa.
> > For those in positions (b): they are loudspeaker, sofa, airplane, and bench.
> > And for positions (c): they are airplane, bench, car, and rifle.
> > The order we're referencing is based on the four corners, starting from the top-left, moving to the top-right, then bottom-left, and finally bottom-right.
> >
> > Please don't hesitate to reach out if you have any further questions.

---

> > > ### Comment · Reviewer_2vix · 2023-08-20
> > >
> > > Thanks for the response and inter-category interpolation results which are promising and demonstrate the effectiveness of the method. I am retaining my score and would be happy to see this paper accepted.

---

> > > > ### Author Response · Authors · 2023-08-20
> > > >
> > > > Thank you for giving our paper a positive score. We are delighted to know that you appreciated the inter-class interpolation results in ShapeNet. We'll include a wider range of interpolation examples in the final version of our work.

---

### Official Review · Reviewer_uPSW · 2023-07-09

**Soundness:** 3 good
**Presentation:** 3 good
**Contribution:** 3 good
**Rating:** 6
**Confidence:** 3

**Summary:**

The paper proposes a mixture of neural implicit functions for generating neural fields via model averaging of implicit representations trained through context adaptation (meta-learning/auto-decoding paradigm) and subsequently task-specific generalization (additional sampling strategy with respect to the latent mixture coefficient vector).

**Strengths:**

This well-written paper clearly elucidates its contributions with convincing results.

**Weaknesses:**

- The authors are requested to discuss limitations within the bounds of the main paper.
- Minor Corrections
    - Line 108, resepctively → respectively
    - Line 226, the both → both the
    - Line 278, effective learn → effectively learns
    - Caption of Figure 1 ideally should elucidate the concept of neural radiance fields.

**Questions:**

Experiments lack quantitative evaluation/baseline. The authors are requested to clarify/highlight baseline models in proposed evaluation protocols.

**Limitations:**

- The proposed approach does not scale.
- The current model exhibits aliasing artifacts in the neural radiance field scene.
- There’s potential for bias within the system.

---

> ### Author Rebuttal · Authors · 2023-08-10
>
> We are glad to hear your positive comment. We will reflect the minor corrections you mentioned in the final version. The other concerns will be answered as below.
>
> > Experiments lack quantitative evaluation/baseline. The authors are requested to clarify/highlight baseline models in proposed evaluation protocols.
>
> Since this research area has actively developed recently, we were struggling to compare our method to the earlier methods.
>
> We adopted the models (i.e., GEM, GASP, and Functa) that simultaneously deal with 2D images and 3D scenes as baselines These methods can be easily trained just by changing the dataset with equivalent architecture for all benchmarks with different modalities.
>
> We follow the protocol from Functa for images and the protocol from GEM for voxels.  because the two models are two-stage generative INR and exploit context adaptation procedures. On evaluation, we used an evaluation protocol from Functa for images and NeRF and GEM for voxels. We will revise the paper to clarify the baselines.

---

> > ### Comment · Reviewer_uPSW · 2023-08-15
> > **Response to Authors**
> >
> > Dear Authors, \
> > Appreciate your response - which addresses the major concern i.e., benchmarking. \
> > However, due to limitations concerning scalability, I hereby retain my original score of 6.

---

> > > ### Author Response · Authors · 2023-08-15
> > >
> > > We are pleased to know that you appreciate our benchmark compared to the baselines over three distinct modalities and are grateful for your positive rating of our work.
> > > We are currently addressing the scalability concerns also mentioned by reviewer ZMK5 and plan to discuss this thoroughly before the end of the discussion period.
> > > Please don't hesitate to reach out if you have any further questions.

---

### Official Review · Reviewer_fjUK · 2023-07-10

**Soundness:** 2 fair
**Presentation:** 4 excellent
**Contribution:** 2 fair
**Rating:** 5
**Confidence:** 5

**Summary:**

This paper introduces an approach for learning generative neural fields using linear combinations of implicit basis networks, called Mixture of Neural Implicit Functions (mNIF). The basis networks and their coefficients in mNIF can be learned through meta-learning or auto-decoding techniques. By this means, the capacity of generative neural fields can be increased by adding more basis networks while maintaining the size of each instance. The optimized network is further fine-tuned to make generalized predictions by optimizing with respect to latent vectors, enabling the representation of unseen data. Experimental results demonstrate performance across diverse benchmarks.

**Strengths:**

+ The paper demonstrates strong writing, with clear notation and intuitively-rendered figures. The proposed method is well-motivated and easy to understand.

+ The experiments, although conducted on simple datasets like CelebA and Shapenet, effectively validate the method's effectiveness and parameter efficiency in both 2D and 3D domains. The proposed method achieves competitive performance in generating Implicit Neural Representations (INRs), surpassing the performance of all compared baselines according to metrics. The interpolation results show promise.

**Weaknesses:**

- Some important related work and baselines are missing, including three papers (references [1][2][3]) that explore the idea of using Mixtures of Experts (MoE) in INRs, as well as reference [4] as an additional image generation baseline and reference [5] as a strong baseline for shape generation. As a result, the proposed method does not appear entirely novel in the field and resembles existing latent code-based implicit representations, differing primarily in the incorporation of latent information through different computation paradigms. The use of MoE is also common in the community. It is recommended to exhaustedly discuss the difference between the aforementioned works and highlight key contributions.

- The motivation for using the mixture of Neural Implicit Fields (NIFs) is mentioned as better memory efficiency, but it is recommended to directly compare run-time memory with the baselines. Additionally, it would be beneficial to include inference time comparisons with the presented methods.


[1] NeurMiPs: Neural Mixture of Planar Experts for View Synthesis

[2] Neural Implicit Dictionary via Mixture-of-Expert Training

[3] Switch-NeRF: Learning Scene Decomposition with Mixture of Experts for Large-scale Neural Radiance Fields

[4] Adversarial Generation of Continuous Images

[5] HyperDiffusion: Generating Implicit Neural Fields with Weight-Space Diffusion

**Questions:**

1. What training method is used throughout the experiments?

2. Why Functa is not compared in the voxel reconstruction/generation task on ShapeNet?

**Limitations:**

The main text of the paper does not address the limitations of the proposed method. From my understanding, scaling up the method to larger and more complex datasets could still present challenges.

---

> ### Author Rebuttal · Authors · 2023-08-10
>
> Thank you for your constructive comments on improving the writing and analysis of the proposed method. We answered your two concerns on missing baselines and analysis on inference cost as below.
>
> ## W1. missing related works and baselines
>
> ### INR with mixture of experts
> Thank you for suggesting the related research. The research you've mentioned is closely related in that it constitutes mixtures of experts. However, there are two main differences between our research and the one you've suggested. Firstly, our research dealt with generative neural fields or code-based implicit representation, whereas the cited study only tackled single scene fitting. The second difference is whether INR is used as an expert or not.
>
> Research on mixtures of experts has two design points: 1) which experts to use and 2) how to combine the results of these experts. The cited studies employ INR as an expert and structure the model by either ensembling or selecting from the outputs of the experts. Our research significantly differs in that we construct the mixture using INR’s weight. This seems like a minor difference, but it showcases a major difference in terms of INR's efficiency.
>
> In the case of [1, 2], calculations must be performed for all experts, so there's no computational advantage. In the case of [3], only the selected expert is used, and the rest are ignored, which offers a computational advantage. However, the full capacity of model isn't utilized at all.
>
> Our study avoids such trade-offs of mixtures of experts by structuring mixtures based on weights of INR. By doing so, the entire model capacity is utilized, and the INR operates with averaged-out weight, thereby promoting inference efficiency. This feature is especially efficient in 3D voxels or NeRF scenes with numerous input coordinate queries. To the best of our knowledge, there is no INR work on aggregating entire model weights within weight space. We will revise the paper to cover discussion referred works.
>
> ### Additional baselines for generative neural fields
> In our study, we focus on algorithms for generating neural fields; especially the algorithms that can handle **every modality such as 2D image, 3D voxel, and NeRF**. That’s why we take GEM, GASP, Functa as the baseline methods for our work.
>
> Contrary to our baselines, INR-GAN [4] is only capable of image generation. INR-GAN relies on an image-specific convolutional discriminator while generator is based on INR. So, this work can not fully take the benefits of continuous representation of INR and. We will add this work as baseline of image-specific INR work in our manuscript with discussion.
>
> HyperDiffusion [5] is concurrent to our work, which will appear in ICCV23. The proposed framework is generic to modality, however, this work demonstrates generation on voxel and voxel video, not on images. We will be able to make a comparison after the release of the detailed implementation and introduced dataset.
>
> ## W2. analysis on inference costs - speed and memory
> Our algorithm is efficient in terms of inference cost compared to our baselines. We measure the memory consumption and inference cost of a single sample (generated INR). Please check the Table A of the attached PDF for detailed information.
>
> GEM / Functa: Our algorithm shows efficient inference in all modalities (image, voxel, NeRF) with better performance. Notably in SRN Cars, our model demonstrates efficient computation and memory over Functa; **199 times less FLOPS, 49 times faster fps, and 22 times less memory** in single view inference. It reinforces our motivation for the proposed method.
>
> GASP: GASP shows more efficient inference compared to our models reported in the main paper. We additionally report the light configuration for our model on image and voxel. Light configuration of our model is the most efficient method in image and voxel and also outperforms the baselines in voxel. In image domain, our light model performs worse than GASP. We conjecture that FID metric over-penalizes blurriness, which is discussed in several works including Functa.
>
> We count FLOPS with the fvcore library [6] and check inference time and memory usage with PyTorch Profiler [7].
>
> ## Q1. training method throughout experiments
> We use two steps for training our generative neural field.
> In the first step, we conduct a context adaptation by meta-learning (algorithm1) or auto-decoding (algorithm2) to training our generative neural field. This procedure train neural implicit functions, or experts, in all mixtures to represent each sample (an image, a voxel, or a NeRF) in the training set with a context vector. As a result, the context vector modulates the neural field to represent a specific sample.
> In the second step, the generative model is trained with the discovered context vectors (of training samples); Denoising diffusion probabilistic process is selected for the generative model. Trained generative model can generate a novel context vector and we can build a neural field with this context vector.
> The advantage of this two-stage methodology is 1) acquiring compact representation for neural fields and 2) saving computational cost during training by decoupling the learning neural fields and the training generative model.
>
> ## Q2. Missing voxel performance of Functa.
> We can not reproduce the performance of Functa due to a longer training time of Functa on voxel generation. Note that Functa requires 25.5 times more computational costs in FLOPS than our model as reported in Table A of the attached PDF. We adopt reported performances from the baselines while Functa does not report reconstruction and generation performances on ShapeNet $64^3$ with evaluation metric introduced in GEM. We will report voxel performance of reproduced Functa with our training setting.
>
> [6] fvcore, https://github.com/facebookresearch/fvcore/blob/main/docs/flop_count.md
> [7] PyTorch Profiler, https://pytorch.org/tutorials/recipes/recipes/profiler_recipe.html

---

> > ### Author Response · Authors · 2023-08-15
> >
> > We strive to provide comprehensive answers and hope our responses have addressed your queries. Nevertheless, we acknowledge that there might be moments of ambiguity or potential misunderstandings. Please don't hesitate to seek further clarification on any aspect of our work.
> >
> > We're also grateful for the points you raised concerning weakness 2. Your insights not only reinforce our motivation but also quantitatively demonstrate the efficiency of our model, particularly in the voxel and NeRF scenes. Here's a summary of the efficiency metrics from Table A in the attached PDF file:
> >
> > CelebA-HQ $64^2$        | Ours (light) | GASP  | Ours (main paper) |GEM   | Functa
> > |--------------------------------|---------------|----------|-------------------------|---------|----------|
> > Inference cost (GFLOPS)  | 0.069     | 0.305  | 0.340 |3.299  | 8.602
> > Inference speed (fps)       | 2841.0  | 1949.3 | 891.3 | 559.6 | 332.9
> > Inference memory (MB)  | 10.16     | 16.35  | 24.42 |70.31 | 144.05
> >
> > ShapeNet $64^3$          | Ours (light)| GASP     | Ours (main paper) |GEM     | Functa
> > |--------------------------------|---------------|----------|--------------------------|---------|----------|
> > Inference cost (GFLOPS)   | 4.363    | 8.742    | 21.613   | 207.0   | 550.0
> > Inference speed (fps)        | 191.5    | 180.9    | 69.6        | 16.7     | 6.3
> > Inference memory (MB)   | 642.07 | 763.13  | 1513.26 | 4010.0  | 9000.0
> >
> > SRN Cars                             | Ours (main paper)   | Functa
> > |--------------------------------|---------------------------|------------|
> > Inference cost (TFLOPS)   | 0.009  | 1.789
> > Inference speed (fps)        | 97.7    | 2.0
> > Inference memory (GB)    | 1.26    | 28.01
> >
> > Our light configuration for images is given by $(M,D,W,H)=(256,4,64,512)$, and for voxels, it's $(M,D,W,H)=(256,4,64,1024)$. Here, $M,D,W,H$ represent the number of mixtures, hidden layers, hidden layer dimensions, and context vector dimensions, respectively.
> >
> > Additionally, to underscore the efficiency of our model during the training process, as requested by Reviewer ZMK5, we've measured memory consumption during meta-training process in context adaptation:
> >
> > Memory		| Ours		| Functa
> > |-------------------------------|--------------------|-------------|
> > Training (batch 32)	| 9.3 GB 	| 40.2 GB
> > Inference 		| 24.4 MB	| 144.1 MB
> >
> > It's worth highlighting that our method utilizes 4.3 times less GPU memory for training and 5.9 times less for inference compared to Functa. While our training phase might slightly exceed our inference memory requirements, the efficiency gap between our method and Functa remains significant. This underlines that the small size of inference parameters can effectively mitigate computational burdens arising from handling large-scale coordinate input in both training and inference.

---

> > > ### Comment · Reviewer_fjUK · 2023-08-20
> > > **Thanks for rebuttal!**
> > >
> > > My major concerns lie in the lack of discussion on the existing works. The authors' rebuttal addresses this problem and now I'm lean toward acceptance. I expect authors to include the discussions above into the camera-ready version.

---

> > > > ### Author Response · Authors · 2023-08-21
> > > >
> > > > Thank you for your insightful feedback during the discussion phases and for improving the rating positively on our work. We will include mentioned comments and discussions in the final version of the paper.

---

### Author Rebuttal · Authors · 2023-08-10

We sincerely appreciate your valuable comments to improve our work. We presume that there are no strong negative comments and we earnestly responded to your concerns; please see the respective comments. Before answering your questions and concerns, we would like to highlight our contributions by quoting your comments.

Firstly, all the reviewers consented that our proposed method is novel and interesting; **Reviewer fjUK** commented *The proposed method is well-motivated and easy to understand,* **Reviewer uPSW** commented *This well-written paper clearly elucidates its contributions with convincing results,* **Reviewer 2vix** commented *The methodology is well explained and is simple but effective,* **Reviewer ZMK5** commented *The observation that the linear combination of neural fields as basis function parameters is quite interesting and novel,* and **Reviewer YZcJ** commented *The proposed model is an interesting direction with a potential to contribute to the research of this topic.*

Secondly, many reviewers left positive comments on the experimental results; **Reviewer fjUK** commented *The experiments, although conducted on simple datasets like CelebA and Shapenet, effectively validate the method's effectiveness and parameter efficiency in both 2D and 3D domains. (skipped) The interpolation results show promise,* **Reviewer uPSW** commented *This well-written paper clearly elucidates its contributions with convincing results,* **Reviewer 2vix** commented *The inference performance is significantly better than three state-of-the-art methods considered, and so are the extensive generation metrics like FID, precision, recall, coverage etc,* and **Reviewer ZMK5** commented *The quantitative results are better than prior works.*

Lastly, most of the reviewers agreed that our paper is well-written and well-organized; **Reviewer fjUK** commented *The paper demonstrates strong writing, with clear notation and intuitively-rendered figures,* **Reviewer uPSW** commented *This well-written paper clearly elucidates its contributions with convincing results,* **Reviewer 2vix** commented *The paper is well-written and easy to follow,* and **Reviewer ZMK5** commented *The paper is generally well-written and easy to follow, while some important details are missing in the current status.*

---

> ### Author Response · Authors · 2023-08-20
>
> We deeply appreciate the insightful reviews and discussions throughout the review period. We will revise our paper to incorporate your valuable feedback. Below, we provide summaries of the reviews, our responses, and discussions.
>
> Our primary objective is to reconcile two seemingly contrasting goals: efficient inference and expansive capacity in an implicit neural representation (INR). Unlike other research on mixtures of INRs, our approach constructs mixtures within the INR's weight space. The computational cost of INR is dictated by the size of inference parameters, not by the total size of learnable parameters. But the sample quality is in favor of the size of the learnable parameter. Ensembling within the weight space maintains the size of the inference parameter compact while allowing us to easily increase learnable parameter size by adding more mixtures. This design allows our model to handle tremendous coordinate inputs with comparable visual quality.
>
> We evaluate the efficiency of our proposed method against INR baselines based on constructive feedback from reviewers fjUK and JMK5. We observe that our approach demonstrates a high degree of efficiency. Especially in NeRF scenes with a significant number of input coordinates, our method achieves 199 times fewer FLOPS, 49 times faster fps, and 22 times reduced memory than Functa during single view inference. Detailed metrics can be found in Table A of the attached file.
>
> SRN Cars                             | Ours   | Functa
> |--------------------------------|---------|-----------|
> Inference speed (fps)       | 97.7    | 2.0
> Inference cost (TFLOPS)   | 0.009  | 1.789
> Inference memory (GB)   | 1.26    | 28.01
> Reconstruction PSNR       | 25.9    | 24.2
> Generation FID                  | 79.5    | 80.3
>
> Reviewers fjUK and uPSW YZcJ raised questions regarding our choice of baselines. We have selected GEM, GASP, and Functa for this purpose. Our model and selected baselines focus on algorithms for generating neural fields, especially those algorithms handling various modalities such as 2D images, 3D voxels, and NeRF scenes. Our model exploits a two-step training process. Initially, we perform context adaptation on our INR model to obtain context vectors for the training set. Subsequently, we train a latent diffusion model for sampling context vectors. The reconstruction performance pertains to the first step, while the generation performance is influenced by both steps. As reviewer YZcJ noted, increasing the latent space's representation size improves reconstruction performance. But, generation performance may require a different optimal representation size.
>
> Upon request by reviewer YZcJ, we delved deeper into our model's internal states. Our observations indicate that information for each sample is evenly distributed across all mixtures within the hidden layers except the input and output layers. The main paper's Figure 5 showcases the diversity of learned weights in each mixture, implying that each mixture's neural implicit function contributes equally to representing a training set sample. Figure A(a) in the attached file further substantiates our model's distributed nature. As the number of mixtures increases, the quality of the reconstructed image steadily improves. It also explains that the information is distributed information over mixtures. Figure A (b) offers a visual representation of the impact of adding or subtracting coefficients from each layer. We can observe that information is predominantly located in the hidden layer, excluding the input and output layers.
>
> In Figure B of the attached file, we visualize the inter-class interpolations requested by reviewer 2vix. We plot three interpolations with four different classes correspondingly. (a) chair (top left), cabinet (top right), display (bottom left), sofa (bottom right) (b) loudspeaker, sofa, airplane, bench (c) airplane, bench, car, rifle. Figure B (c) shows smooth interpolation even with distinct shapes.
>
> We have one more day for discussions related to this work. We invite you to review our findings and share any further queries or concerns.
>
> Thank you for your time and feedback.

---

### Decision · Program_Chairs · 2023-09-21

**Decision:**

Accept (poster)

**Comment:**

Author rebuttal addressed most of the reviewer concerns and all reviewers are leaning accept after final discussion.